# HSIC Bottleneck for Cross-Generator and Domain-Incremental Synthetic Image Detection

**Chin-Chia Yang**[1,4]**, Yung-Yu Chuang**[1,2]**, Hwann-Tzong Chen**[3] **& Tyng-Luh Liu**[4]
[1]Department of Computer Science, National Taiwan University, Taiwan
[2]AI Center of Research Excellence (AI-CoRE), National Taiwan University, Taiwan
[3]Department of Computer Science, National Tsing Hua University, Taiwan
[4]Institute of Information Science, Academia Sinica, Taiwan
`{f10942093,cyy}@csie.ntu.edu.tw`
`htchen@cs.nthu.edu.tw,liutyng@iis.sinica.edu.tw`

## Abstract

Synthetic image generators evolve rapidly, challenging detectors to generalize across current methods and adapt to new ones. We study domain-incremental synthetic image detection with a two-phase evaluation. Phase I trains on either diffusion- or GAN-based data and tests on the combined group to quantify bidirectional cross-generator transfer. Phase II sequentially introduces renders from 3D Gaussian Splatting (3DGS) head avatar pipelines, requiring adaptation while preserving earlier performance. We observe that CLIP-based detectors inherit text-image alignment semantics that are irrelevant to authenticity and hinder generalization. We introduce a Hilbert-Schmidt Independence Criterion (HSIC) bottleneck loss on intermediate CLIP ViT features, encouraging representations predictive of real versus synthetic while independent of generator identity and caption alignment. For domain-incremental learning, we propose HSIC-Guided Replay (HGR), which selects per-class exemplars via a hybrid score combining HSIC relevance with k-center coverage, yielding compact memories that mitigate forgetting. Empirically, the HSIC bottleneck improves transfer between diffusion and GAN families, and HGR sustains prior accuracy while adapting to 3DGS renders. These results underscore the value of information-theoretic feature shaping and principled replay for resilient detection under shifting generative regimes.

## 1 Introduction

The rapid progress of generative models has led to increasingly realistic synthetic images, raising urgent concerns about the spread of misleading digital content. The detection problem is inherently open-world: new diffusion architectures, GAN variants, and 3D Gaussian Splatting (3DGS) rendering pipelines will continue to emerge, remaining unseen during training. Among these, 3DGS has enabled photorealistic, real-time head avatars, expanding the scope of rendered imagery beyond traditional 2D synthesis. As illustrated in Figure 1, synthetic images produced by GANs, Deepfake, and 3DGS exhibit distinct artifacts and statistical patterns, motivating detectors that generalize beyond any single generative family.

A practical detector therefore requires two key capabilities: (i) robust generalization across diverse generation paradigms and (ii) continual adaptability to incorporate new synthetic sources without catastrophic forgetting. Existing systems that rely on the vision backbone of CLIP show promising cross-generator transfer between diffusion and GAN data. However, CLIP embeddings are primarily optimized for text-image alignment, embedding caption semantics that are irrelevant to authenticity and potentially detrimental when the task is purely image-based. As shown in Figure 2, the raw CLIP features cluster together according to object class rather than the real-synthetic boundary.

We tackle these challenges with two complementary components integrated into a single pipeline. First, we introduce a Hilbert-Schmidt Independence Criterion (HSIC) bottleneck on features ex-

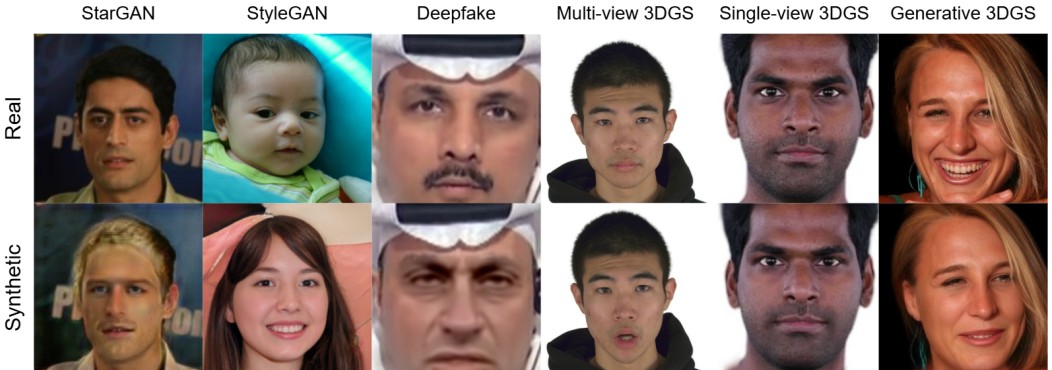

Figure 1: **Diversity of synthetic images across paradigms**. Columns show representative sources of synthetic faces: StarGAN, StyleGAN, Deepfake (face swap), multi-view 3D Gaussian Splatting (3DGS) head avatar, single-view 3DGS head avatar, and generative 3DGS head avatar. Top row: real images. Bottom row: corresponding synthetic examples. The breadth of GAN-based synthesis and rendered 3DGS avatars creates substantial distribution shifts, underscoring the need for detectors that both generalize to unseen sources and continually adapt as new synthetic sources emerge.

tracted from the intermediate layers of the CLIP ViT. Specifically, we aggregate multi-layer representations, project them to a compact latent space, and optimize an HSIC objective that minimizes dependence on the input while maximizing dependence on the label. This regularization suppresses nuisance factors (including text-alignment signals) and produces generator-invariant yet discriminative representations, strengthening mutual generalization between diffusion and GAN models.

Second, for continual learning on rendered domains, we propose HSIC-Guided Replay (HGR). During adaptation, HGR constructs a compact exemplar memory per class by ranking candidates with a weighted score that combines HSIC relevance (information centrality) and k-center coverage (spatial spread). These exemplars are replayed alongside new data to mitigate forgetting and stabilize performance across evolving domains. To support this setting, we curate three 3DGS head avatar datasets covering multi-view reconstruction, single-view reconstruction, and a generative pipeline. Each dataset provides paired real and synthetic frames with identity-disjoint splits and standardized preprocessing. The synthetic images are highly challenging and cannot be reliably detected by existing synthetic image detectors. Our evaluation first trains detectors exclusively on diffusion or GAN images to measure cross-generator transfer, and then sequentially adapts them to the curated 3DGS domains under all adaptation orderings, while continuously monitoring prior-domain accuracy.

In summary, our contributions are threefold.

- **HSIC Bottleneck for Generalized Synthetic Image Detection**. We impose an HSIC bottleneck on intermediate CLIP features to suppress text-alignment nuisances and amplify image-label dependence, substantially improving cross-generator generalization.

- **HSIC-Guided Replay (HGR) for Continual Adaptation**. We introduce an HSIC-driven exemplar selection and weighting scheme that delivers compact yet effective replay, enabling adaptation to 3DGS content while preserving prior accuracy.

- **3DGS Synthetic Image Benchmark**. We curate a 3DGS rendered image suite spanning multi-view reconstruction, single-view reconstruction, and a generative 3DGS pipeline, offering a benchmark to advance research on synthetic image detection.

To the best of our knowledge, we are the first to introduce a photorealistic 3DGS Synthetic Image Benchmark, together with a continual learning solution tailored to this setting, providing the community with a much-needed, realistic testbed for studying robustness and generalization of synthetic image detectors under evolving 3D generation pipelines.

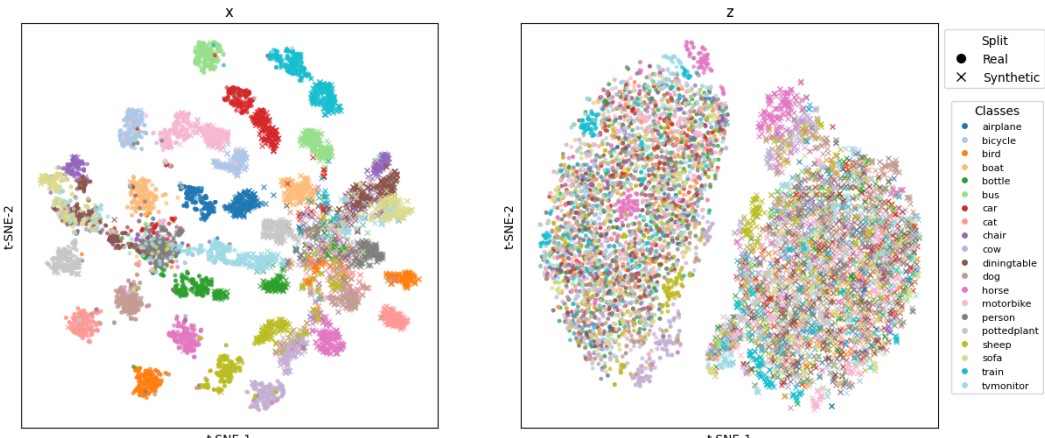

Figure 2: **HSIC bottleneck transforms CLIP semantic clusters into real/synthetic separability**. t-SNE visualization of features before ($x$, left) and after ($z$, right) applying the HSIC bottleneck. Points are colored by semantic class, with markers denoting Real (•) and Synthetic (×). Pretrained CLIP features $x$ mainly cluster by object category, intermixing real and synthetic samples within each class. By contrast, the HSIC bottleneck suppresses nuisance semantics and reshapes the representation to align with the labels, causing real images to cluster together and synthetic images to cluster together across categories, thereby producing a clearer decision boundary for detection.

## 2 RELATED WORK

### 2.1 GENERALIZED SYNTHETIC IMAGE DETECTION AND 3DGS HEAD AVATARS

Wang et al. (2020) first explicitly posed the problem of generalization to unseen generators in synthetic image detection, showing that with careful pre/post-processing and augmentation, a classifier trained on one GAN (e.g., ProGAN) can transfer to other generators, indicating shared synthesis artifacts. Following Wang et al. (2020), many works have further explored techniques that identify forensic cues for detecting synthetic images (Qian et al., 2020; Chai et al., 2020). LGrad (Tan et al., 2023) moved toward generator-agnostic cues by operating in gradient space, yielding strong cross-generator generalization for GAN fakes. NPR (Tan et al., 2024) rethinks source-invariant artifacts by analyzing up-sampling operations common to GANs and diffusion pipelines, proposing neighboring pixel relationships as a simple, local pixel-dependency descriptor that generalizes across a wide range of generators.

Recent work has begun to specifically target diffusion-generated image detection. DIRE (Wang et al., 2023a) leverages diffusion inversion–based reconstruction to expose discrepancies between real and synthetic images, using reconstruction error patterns as robust cues for detecting diffusion-based fakes. Building on this idea, DRCT (Chen et al., 2024) further exploits reconstruction signals through a contrastive objective: real and reconstructed pairs are pulled together, while diffusion-generated and reconstructed pairs are pushed apart. This reconstruction-aware contrastive training encourages the detector to capture generator-agnostic artifacts from the diffusion process, leading to strong cross-model and cross-dataset generalization.

A complementary line of work leverages large vision–language backbones (Radford et al., 2021; Oquab et al., 2024; Touvron et al., 2023), specifically CLIP (Radford et al., 2021), for generalized synthetic image detection (Ojha et al., 2023; Koutlis & Papadopoulos, 2024; Zhang et al., 2025; Yang et al., 2025). UniFD (Ojha et al., 2023) demonstrated that operating directly in frozen CLIP feature space—using a nearest-neighbor or a linear probe—provides markedly improved transfer to unseen families, including mutual transfer between diffusion and GAN models. Building on this idea, RINE (Koutlis & Papadopoulos, 2024) extracts intermediate encoder-block representations from CLIP rather than only the final layer, capturing richer structural and semantic cues that enhance out-of-distribution generalization. Most recently, VIB-Net (Zhang et al., 2025) couples a pretrained CLIP backbone with a variational information bottleneck to suppress task-irrelevant factors

while retaining discriminative evidence, pushing universal detection performance across generator families.

Beyond diffusion and GAN imagery, 3D Gaussian Splatting (3DGS) enables photorealistic, real-time rendering with explicit point-based primitives and has rapidly become a foundation for head avatar synthesis, introducing rendered-fake sources that differ from conventional image synthesis. We highlight three representative families that define our rendered-fake domains: *(i) Multi-view 3DGS head avatar*—Gaussian Head Avatar (Xu et al., 2024) proposes dynamic, controllable 3D Gaussians for ultra high-fidelity head modeling, pairing a learned deformation field with an explicit 3DGS representation and multi-view supervision; *(ii) Single-view 3DGS head avatar*—SplattingAvatar (Shao et al., 2024) embeds Gaussians on a deformable mesh to disentangle motion and appearance, supporting real-time rendering from monocular training signals; and *(iii) Generative 3DGS head avatar*—GAGAvatar (Chu & Harada, 2024) predicts 3DGS parameters from a single image for one-shot, animatable, and generalizable avatars. These three categories define the rendered-fake domains in our continual learning protocol, complementing diffusion and GAN synthesis, and support systematic cross-generator evaluation. Despite the progress of generalized detection on diffusion and GAN, detectors that transfer well across these paradigms frequently falter on 3DGS-rendered fakes, motivating methods that couple stronger generalization with principled continual adaptation.

## 2.2 CONTINUAL LEARNING

We investigate continual learning in the domain-incremental regime: the label space remains fixed, while the input distribution shifts across domains—first among diffusion or GAN generators, and later to 3DGS categories. Continual methods are commonly grouped into three families: *regularization-based*, *architecture-based*, and *rehearsal-based*. Regularization-based approaches (Kirkpatrick et al., 2017; Zenke et al., 2017; Li & Hoiem, 2018; Aljundi et al., 2018) constrain parameter updates to protect knowledge from earlier domains. Architecture-based methods (Rusu et al., 2016; Mallya & Lazebnik, 2018; Yan et al., 2021; Wang et al., 2025) expand capacity or allocate disjoint subnetworks to reduce interference. Rehearsal-based methods maintain a compact replay memory and interleave past exemplars with current data. *Class-mean herding* (iCaRL) (Rebuffi et al., 2017) selects exemplars near class centroids to ensure representativeness. *Class-balanced reservoir sampling* (CBRS) (Chrysakis & Pourkamali-Anaraki, 2020) adapts classical reservoir sampling to preserve label balance in nonstationary streams. Coverage-oriented selection via the greedy *k-center* (farthest-first) heuristic (Gonzalez, 1985; Sener & Savarese, 2018) spreads exemplars across the feature space to improve diversity and reduce redundancy. DualH-SIC (Wang et al., 2023b) leverages the Hilbert–Schmidt Independence Criterion (HSIC) to explicitly model inter-task relationships. It introduces two complementary losses: HSIC-Bottleneck for Rehearsal (HBR), which penalizes dependence between buffered inputs and intermediate features while preserving label–feature dependence to suppress task-specific nuisance information, and HSIC Alignment (HA), which maximizes dependence between current-task and buffered features to promote task-invariant knowledge sharing. Since replay is arguably the most widely adopted and practically deployed mechanism in continual learning, we focus on what we view as its key factor: the sampling strategy. As the first work to study a continual learning setup for synthetic image detection, we build on this rehearsal-based line of work in the domain-incremental setting and introduce HSIC-Guided Replay, a replay-based sampling strategy tailored to synthetic image detection that scores and selects exemplars to preserve coverage of prior domains while adapting to new ones.

## 2.3 HILBERT-SCHMIDT INDEPENDENCE CRITERION (HSIC)

HSIC (Gretton et al., 2005) is a kernel-based measure of statistical dependence between random variables, defined via reproducing kernel Hilbert spaces (RKHSs). Let $\mathbf{a} \in \mathbb{A}$ and $\mathbf{b} \in \mathbb{B}$ be random variables associated with RKHSs $(\mathbb{F}, k)$ and $(\mathbb{G}, \ell)$, induced by feature maps $\phi : \mathbb{A} \to \mathbb{F}$ and $\psi : \mathbb{B} \to \mathbb{G}$, respectively. Let the corresponding mean embeddings be $\boldsymbol{\mu}_{\mathbf{a}} = \mathbb{E}[\phi(\mathbf{a})]$ and $\boldsymbol{\mu}_{\mathbf{b}} = \mathbb{E}[\psi(\mathbf{b})]$. The cross-covariance operator $\mathcal{C}_{\mathbf{ab}} : \mathbb{G} \to \mathbb{F}$ is defined as

$$\mathcal{C}_{\mathbf{ab}} = \mathbb{E}\big[(\phi(\mathbf{a}) - \boldsymbol{\mu}_{\mathbf{a}}) \otimes (\psi(\mathbf{b}) - \boldsymbol{\mu}_{\mathbf{b}})\big]. \tag{1}$$

The population HSIC is the squared Hilbert–Schmidt norm of this operator:

$$\mathrm{HSIC}(P_{\mathbf{ab}}, \mathbb{F}, \mathbb{G}) = \big\|\mathcal{C}_{\mathbf{ab}}\big\|_{\mathrm{HS}}^{2}. \tag{2}$$

**Kernel expectation form (population).** Let $(\mathbf{a}', \mathbf{b}')$ be an independent copy of $(\mathbf{a}, \mathbf{b})$. Expanding equation 2 yields the following equivalent expression in terms of kernels $k$ and $\ell$:

$$\text{HSIC}(\mathbf{a}, \mathbf{b}) = \mathbb{E}_{\mathbf{aa'bb'}}\big[k(\mathbf{a}, \mathbf{a}')\,\ell(\mathbf{b}, \mathbf{b}')\big] + \mathbb{E}_{\mathbf{aa'}}\big[k(\mathbf{a}, \mathbf{a}')\big]\,\mathbb{E}_{\mathbf{bb'}}\big[\ell(\mathbf{b}, \mathbf{b}')\big]$$
$$- 2\,\mathbb{E}_{\mathbf{ab}}\Big[\mathbb{E}_{\mathbf{a'}}k(\mathbf{a}, \mathbf{a}')\,\mathbb{E}_{\mathbf{b'}}\ell(\mathbf{b}, \mathbf{b}')\Big], \tag{3}$$

which is zero if and only if $\mathbf{a}$ and $\mathbf{b}$ are independent under suitable conditions, e.g., when characteristic kernels are used.

**Empirical estimator.** Given $n$ i.i.d. samples $\{(a_i, b_i)\}_{i=1}^n$ from $P_{\mathbf{ab}}$, define the Gram matrices $\mathbf{K}, \mathbf{L} \in \mathbb{R}^{n \times n}$ by $\mathbf{K}_{ij} = k(a_i, a_j)$ and $\mathbf{L}_{ij} = \ell(b_i, b_j)$. Let $\mathbf{1} \in \mathbb{R}^n$ denote the all-ones vector, $\mathbf{I}_n \in \mathbb{R}^{n \times n}$ the identity matrix, and the centering matrix $\mathbf{H} = \mathbf{I}_n - \frac{1}{n}\mathbf{1}\mathbf{1}^\top$, which is symmetric and idempotent. A commonly used (biased) V-statistic estimator is

$$\widehat{\text{HSIC}}(\mathbf{a}, \mathbf{b}) = \frac{1}{(n-1)^2}\,\text{tr}\big(\mathbf{KHLH}\big) = \frac{1}{(n-1)^2}\,\text{tr}\big(\mathbf{HKHHLH}\big) = \frac{1}{(n-1)^2}\,\text{tr}\big(\bar{\mathbf{K}}\bar{\mathbf{L}}\big), \tag{4}$$

where $\text{tr}(\cdot)$ denotes the standard matrix trace operator. This estimator provides an efficient empirical estimate without requiring density models. In practice, Gaussian RBF kernels are often adopted for $k$ and $\ell$, and the bandwidth can be set by the median heuristic. The centered versions $\bar{\mathbf{K}} = \mathbf{HKH}$ and $\bar{\mathbf{L}} = \mathbf{HLH}$ correspond to centering the associated feature maps, so that equation 4 matches the population definition equation 2 through equation 3.

## 3 METHOD

Our detector builds on CLIP ViT features, which, like most pretrained extractors, are not tailored to synthetic image detection and often entangle authenticity with generator identity and text-image semantics. To mitigate this, we introduce an HSIC bottleneck that refines features for real-versus-synthetic discrimination while suppressing spurious dependencies. For domain-incremental learning, we further propose HSIC-Guided Replay (HGR), which selects compact exemplars to balance adaptation to new generators with retention of prior knowledge.

### 3.1 HSIC BOTTLENECK

Let the model be $h_\theta = g_{\theta_g} \circ f_{\theta_f}$, where $f_{\theta_f}$ is an encoder and $g_{\theta_g}$ is a classifier. In DualHSIC (Wang et al., 2023b), the encoder is a ResNet with $L$ intermediate layers. Given an input representation $x$, label $y$, and layer-$j$ feature $Z_j$ ($j = 1, \ldots, L$), the layer-wise HSIC objective is

$$\mathcal{L}_{\text{HB}}(\theta_f) = \lambda_x \sum_{j=1}^L \widehat{\text{HSIC}}(x, Z_j) - \lambda_y \sum_{j=1}^L \widehat{\text{HSIC}}(y, Z_j), \tag{5}$$

where $\lambda_x$ encourages compression of input information, while $\lambda_y$ encourages dependence on the label $y \in \{0, 1\}$, indicating whether the sample is synthetic (1) or authentic (0).

Unlike DualHSIC, which applies HSIC at all intermediate layers, we use CLIP ViT as the feature extractor. For each image, we form a CLIP feature representation $x$ by concatenating features from its 24 intermediate layers and the final layer, and then compress it into $z = f_{\theta_f}(x)$. Adapting equation 5 to this setting gives

$$\mathcal{L}_{\text{HSIC-Bottleneck}}(\theta_f) = \lambda_x \widehat{\text{HSIC}}(x, z) - \lambda_y \widehat{\text{HSIC}}(y, z). \tag{6}$$

### 3.2 TRAINING OBJECTIVE

For each sample $i$, let $x_i$ be its CLIP feature representation and $z_i = f_{\theta_f}(x_i)$ be the compressed feature. The classifier $g_{\theta_g}$ outputs a logit $u_i = g_{\theta_g}(z_i)$, and the corresponding probability is $p_i = \sigma(u_i)$, where $\sigma(\cdot)$ is the sigmoid. For binary labels $y_i \in \{0, 1\}$, we use binary cross-entropy:

$$\mathcal{L}_{\text{BCE}}(\theta_f, \theta_g) = -\frac{1}{n}\sum_{i=1}^n \Big[ y_i \log p_i + (1 - y_i)\log\big(1 - p_i\big) \Big]. \tag{7}$$

The final objective is

$$\mathcal{L}_{\text{total}}(\theta_f, \theta_g) = \mathcal{L}_{\text{HSIC-Bottleneck}}(\theta_f) + \mathcal{L}_{\text{BCE}}(\theta_f, \theta_g). \tag{8}$$

### 3.3 HSIC-Guided Replay (HGR)

We combine an HSIC-inspired relevance score with a $k$-center coverage term to select exemplars for the rehearsal buffer. A nonnegative weight $\lambda_{\mathrm{kc}} \geq 0$ controls the strength of the $k$-center regularizer: $\lambda_{\mathrm{kc}}{=}0$ yields pure relevance-based selection, while larger values emphasize coverage. Selection is performed per class $c \in \{0,1\}$, but we omit $c$ below for simplicity. Let $\mathcal{X} = \{x_i\}_{i \in \mathcal{I}}$ be the candidate set of CLIP feature representations with index set $\mathcal{I}$. At step $t$, let $S_{t-1} \subset \mathcal{I}$ be the selected indices (with $S_0 = \varnothing$), and define the active set $A_t = \mathcal{I} \setminus S_{t-1}$.

For each $x_i \in \mathcal{X}$, compute $z_i = f_{\theta_f}(x_i)$. Form the Gaussian RBF Gram matrix $\boldsymbol{K}$ on $\{z_i\}_{i \in \mathcal{I}}$ and let $\bar{\boldsymbol{K}}$ be its centered version as in equation 4. The HSIC-inspired relevance score for index $i \in \mathcal{I}$ is

$$r_i = \left\| \bar{\boldsymbol{K}}_{i,:} \right\|_2^2. \tag{9}$$

To promote coverage and reduce redundancy, we add a $k$-center term in feature space, following the coreset view of Sener & Savarese (2018). Define, for $i \in A_t$,

$$d_i(t) = \begin{cases} \|z_i - \mu\|_2^2, & t{=}1, \\ \min\limits_{j \in S_{t-1}} \|z_i - z_j\|_2^2, & t \geq 2, \end{cases} \quad \text{with} \quad \mu = \frac{1}{|\mathcal{X}|} \sum_{j \in \mathcal{I}} z_j,$$

so that larger $d_i(t)$ favors points farther from the already selected set.

**Selection rule.** HGR selects exemplars by minimizing the $\lambda_{\mathrm{kc}}$-regularized score

$$s_i(t) = \left(1 - \mathcal{N}(r_i)\right) + \lambda_{\mathrm{kc}} \left(1 - \mathcal{N}(d_i(t))\right), \qquad i \in A_t. \tag{10}$$

Here, $\mathcal{N}(\cdot)$ denotes a normalization operation. We choose $i_t^\star = \arg\min_{i \in A_t} s_i(t)$ and update $S_t = S_{t-1} \cup \{i_t^\star\}$ until $|S_t| = m_c$, where $m_c$ is the number of exemplars allocated to class $c$. Repeating this for $c \in \{0,1\}$ and taking the union across classes and domains yields the replay buffer. Intuitively, HGR favors items that are both highly ranked by the relevance score (large $r_i$) and provide coverage (large $d_i(t)$).

## 4 Experiments

### 4.1 Experimental Setup

**Task and protocol.** We study cross-generator synthetic-image detection in two phases under the following protocol. (1) *Cross-generator generalization*: train a detector on one paradigm (diffusion or GAN) and evaluate on the union of diffusion and GAN targets to measure generalization ability. (2) *Domain-incremental learning*: starting from the model in phase (1), continue training as rendered-fakes from 3DGS head avatar pipelines are introduced sequentially (GHA, SA, GAGA-vatar). During this phase, we always monitor performance on previously seen diffusion and GAN test sets to quantify retention alongside adaptation.

**Baselines.** We benchmark against classical and recent detectors for cross-generator generalization, including CNNSpot (Wang et al., 2020), LGrad (Tan et al., 2023), UniFD (Ojha et al., 2023), NPR (Tan et al., 2024), RINE (Koutlis & Papadopoulos, 2024), and VIB-Net (Zhang et al., 2025). For domain-incremental learning, we compare HSIC-Guided Replay with iCaRL (Rebuffi et al., 2017) and CBRS (Chrysakis & Pourkamali-Anaraki, 2020). All rehearsal-based methods share the same per-class memory budget, and all other training hyperparameters are identical across methods.

**Datasets.** For *cross-generator generalization*, we use **GenImage**, which contains diffusion-generated images from Stable Diffusion v1.4/v1.5 (Rombach et al., 2022), ADM (Dhariwal & Nichol, 2021), GLIDE (Nichol et al., 2022), Midjourney (Midjourney, 2022), Wukong (Wukong, 2022), and VQDM (Gu et al., 2022). The GAN sources are taken from the collection of Wang et al. (2020), including ProGAN (Karras et al., 2018), CycleGAN (Zhu et al., 2017), BigGAN (Brock et al., 2019), StyleGAN (Karras et al., 2019), StarGAN (Choi et al., 2018), GauGAN (Park et al., 2019), as well as Deepfake (Rossler et al., 2019) and SAN (Dai et al., 2019).

Table 1: **Cross-generator generalization with diffusion-trained detectors (ACC/AP)**. Each cell reports ACC/AP (%). Within each dataset row, the highest, second highest, and third highest ACC are shaded red, orange, and yellow, respectively. The SDV1.4 row label cell is shaded green to indicate the diffusion training source.

| Dataset | Method (ACC/AP %) | | | | | | | |
|---|---|---|---|---|---|---|---|---|
| | CNNSpot | LGrad | UniFD | NPR | RINE | VIB-Net | Ours | Ours w/ intermediate |
| SDV1.4 | 99.48/99.98 | 99.12/99.94 | 83.55/96.04 | 100.00/100.00 | 96.10/100.00 | 99.55/100.00 | 99.33/99.98 | 99.92/100.00 |
| SDV1.5 | 99.35/99.83 | 99.05/99.92 | 84.80/96.26 | 99.90/99.97 | 95.90/99.90 | 99.20/99.97 | 99.24/99.95 | 99.81/100.00 |
| ADM | 50.10/51.10 | 53.00/58.52 | 53.35/66.34 | 73.00/94.70 | 53.70/86.80 | 73.85/95.49 | 85.82/97.69 | 91.10/99.61 |
| GLIDE | 50.90/58.80 | 64.24/84.00 | 75.30/93.73 | 89.70/95.80 | 56.70/95.70 | 74.25/97.13 | 94.97/99.26 | 97.07/99.85 |
| Midjourney | 56.42/67.93 | 76.34/91.06 | 71.60/92.08 | 82.30/95.50 | 59.20/97.00 | 88.05/97.81 | 83.12/97.35 | 80.86/99.34 |
| Wukong | 97.90/99.80 | 97.53/99.72 | 73.55/90.98 | 100.00/100.00 | 85.00/99.80 | 98.25/99.93 | 98.62/99.92 | 99.86/100.00 |
| VQDM | 50.04/49.92 | 50.93/56.34 | 55.10/74.53 | 68.30/86.30 | 57.40/96.50 | 89.35/97.00 | 93.69/99.25 | 99.42/99.99 |
| ProGAN | 50.27/53.15 | 61.61/83.59 | 58.65/51.77 | 60.30/83.30 | 68.90/81.40 | 89.70/96.59 | 97.54/99.64 | 99.49/99.99 |
| CycleGAN | 49.81/50.23 | 60.74/90.24 | 59.30/63.42 | 67.20/94.90 | 66.50/98.48 | 88.60/98.44 | 97.92/99.86 | 99.66/99.97 |
| BigGAN | 50.10/49.79 | 48.82/47.51 | 61.45/75.81 | 59.20/72.00 | 52.40/94.00 | 91.20/97.17 | 90.28/98.01 | 91.75/99.74 |
| StyleGAN | 50.98/55.98 | 61.43/82.74 | 56.80/54.12 | 58.00/82.70 | 53.30/71.60 | 74.10/84.31 | 94.42/98.93 | 88.15/97.89 |
| StarGAN | 49.77/47.07 | 50.17/99.19 | 61.45/54.93 | 73.20/97.30 | 59.20/97.60 | 80.70/97.60 | 96.05/99.52 | 100.00/100.00 |
| GauGAN | 50.38/56.08 | 49.70/49.25 | 55.30/65.99 | 52.00/66.00 | 51.80/87.90 | 87.15/96.94 | 87.38/94.51 | 90.02/97.60 |
| Deepfake | 51.98/54.86 | 50.17/66.49 | 58.40/70.24 | 74.80/85.30 | 51.60/80.30 | 72.00/81.32 | 66.35/76.47 | 82.15/93.82 |
| SAN | 50.22/54.03 | 56.49/65.09 | 72.00/83.34 | 89.60/95.90 | 62.30/88.70 | 81.50/93.27 | 90.64/96.13 | 88.58/95.51 |
| Avg | 60.51/63.24 | 65.29/78.24 | 65.37/75.31 | 76.50/89.98 | 64.59/91.75 | 85.83/95.53 | 91.69/97.10 | 93.86/98.89 |

For *domain-incremental learning*, we curate a three-part 3DGS benchmark: (i) Gaussian Head Avatar (Xu et al., 2024) (GHA) trained on NeRSemble (Kirschstein et al., 2023) with identity-disjoint, balanced splits (train: 45,772 real / 45,772 synthetic; val: 9,480 / 9,480; test: 9,782 / 9,782); (ii) SplattingAvatar (Shao et al., 2024) (SA) trained on subjects from NeRFace (Gafni et al., 2021), NHA (Grassal et al., 2022), and IM Avatar (Zheng et al., 2022) (train: 20,322 real / 20,094 synthetic; val: 4,007 / 4,036; test: 5,631 / 5,622); and (iii) GAGAvatar (Chu & Harada, 2024), a one-shot generative 3DGS avatar using FFHQ (Karras et al., 2019) inputs with pose-driven reenactment (train: 55,963 real / 55,963 synthetic; val: 6,995 / 6,995; test: 6,996 / 6,996). In this paper, we focus on the 3-domain incremental setting, as evaluating longer domain sequences (e.g., 5+ domains) under multiple orders would require a factorial number of runs and be computationally prohibitive. We prioritize 3DGS-rendered images over NeRF-rendered ones because, in our experience, 3DGS typically produces more photorealistic results.

## 4.2 Cross-Generator Generalization

We study out-of-source transfer by training detectors on a single family (diffusion or GAN) and evaluating across the union of all targets. This setting measures whether a detector trained on one paradigm can generalize to others without exposure to its data.

**Training on diffusion.** Table 1 reports cross-generator generalization with diffusion-only training. Specifically, the detectors are trained on the SDV1.4 dataset. Using intermediate ViT features with an HSIC bottleneck yields the strongest overall mean, outperforming both strong baselines and our non-intermediate variant. Improvements are especially robust on GAN targets and remain consistent across several diffusion datasets, while AP is near-saturated for most rows, indicating stable ranking quality. Although the non-intermediate variant occasionally tops individual entries (e.g., StyleGAN, SAN), the intermediate configuration is overall more consistent and generator-invariant, preserving performance across diverse generators and manipulations.

**Training on GAN.** Table 2 summarizes cross-generator generalization under GAN-only training. Specifically, the detectors are trained on the ProGAN dataset. Using intermediate ViT features with an HSIC bottleneck achieves the best overall mean, outperforming strong baselines while markedly improving transfer to diffusion models. The intermediate configuration secures the best ACC on most diffusion targets and on a subset of GANs (e.g., ProGAN, StarGAN), while remaining competitive elsewhere. AP is near-saturated across large portions of the table, indicating that suppressing nuisance variation at intermediate layers yields a representation that generalizes across generation paradigms without sacrificing ranking quality on the source domain.

Table 2: **Cross-generator generalization with GAN-trained detectors (ACC/AP)**. Each cell reports ACC/AP (%). Within each dataset row, the highest, second highest, and third highest ACC are shaded red, orange, and yellow, respectively. The ProGAN row label cell is shaded green to indicate the GAN training source.

| Dataset | Method (ACC/AP %) | | | | | | | |
| --- | --- | --- | --- | --- | --- | --- | --- | --- |
| | CNNSpot | LGrad | UniFD | NPR | RINE | VIB-Net | Ours | Ours w/ intermediate |
| ProGAN | 99.99/99.99 | 99.80/99.90 | 99.90/100.00 | 99.80/100.00 | 100.00/100.00 | 99.99/100.00 | 99.83/100.00 | 100.00/100.00 |
| CycleGAN | 87.59/96.40 | 86.94/94.01 | 98.50/99.21 | 96.10/98.50 | 99.32/99.99 | 99.00/99.80 | 88.38/99.74 | 93.07/99.99 |
| BigGAN | 71.18/87.50 | 85.63/90.75 | 94.50/98.31 | 84.40/87.80 | 99.60/99.94 | 95.75/99.29 | 90.08/99.53 | 82.05/99.94 |
| StyleGAN | 89.95/96.94 | 91.08/99.80 | 84.40/97.98 | 97.70/99.80 | 88.86/99.44 | 91.25/98.79 | 91.37/98.33 | 93.41/100.00 |
| StarGAN | 94.60/94.24 | 99.27/99.98 | 95.85/99.35 | 99.30/99.90 | 99.55/100.00 | 98.95/99.72 | 97.45/99.87 | 100.00/100.00 |
| GauGAN | 81.44/98.28 | 72.49/79.29 | 99.50/99.80 | 82.50/85.50 | 99.77/100.00 | 99.70/99.99 | 82.32/99.99 | 68.33/99.98 |
| Deepfake | 51.69/64.42 | 56.42/71.71 | 67.40/82.04 | 80.20/82.40 | 80.57/97.90 | 83.20/92.64 | 83.98/92.97 | 82.48/96.82 |
| SAN | 50.00/55.89 | 44.47/45.09 | 56.50/82.18 | 69.20/71.60 | 68.26/94.93 | 70.50/91.62 | 86.99/92.25 | 93.61/97.94 |
| SDV1.4 | 50.82/52.86 | 63.03/70.90 | 63.10/85.48 | 76.60/84.00 | 83.96/98.35 | 71.55/87.24 | 85.79/92.25 | 98.82/99.94 |
| SDV1.5 | 50.88/53.25 | 63.67/71.72 | 63.57/82.30 | 77.90/84.60 | 83.35/98.33 | 70.00/86.98 | 85.14/92.23 | 98.67/99.85 |
| ADM | 60.20/65.14 | 67.10/71.83 | 66.90/84.34 | 69.70/74.60 | 74.61/96.23 | 71.45/87.88 | 81.46/89.61 | 93.01/97.70 |
| GLIDE | 57.85/68.10 | 66.10/75.96 | 61.70/84.04 | 77.30/85.70 | 80.72/97.87 | 69.40/88.53 | 89.72/96.39 | 97.02/99.56 |
| Midjourney | 50.77/56.60 | 56.20/71.42 | 57.85/69.10 | 77.80/85.40 | 57.12/87.41 | 61.25/75.68 | 60.36/66.83 | 69.40/82.49 |
| Wukong | 51.13/51.15 | 63.60/66.51 | 71.06/90.13 | 76.10/80.50 | 84.95/98.62 | 75.90/90.92 | 88.36/95.38 | 98.62/99.88 |
| VQDM | 56.20/69.49 | 67.02/70.23 | 85.00/94.96 | 78.10/81.20 | 89.79/99.23 | 86.65/96.51 | 88.72/96.08 | 97.49/99.74 |
| Avg | 66.95/74.02 | 72.19/78.61 | 77.72/89.95 | 82.84/86.77 | 86.03/97.88 | 82.97/93.04 | 86.66/94.10 | 91.07/98.25 |

Table 3: **Training from SDV1.4**. Cells show **mACC/mAP** (%). In our setup, *base* trains only on SDV1.4 without 3DGS; *Oracle* is jointly trained on SDV1.4 plus {GHA, SA, GAGAvatar} in a non-continual manner; and *iCaRL*, *CBRS*, and our *HGR* are sampling methods for the replay buffer. **Bold** highlights the best mACC among the sampling methods only.

| Method | Diffusion | GANs | Others | GHA | SA | GAGAvatar | Average |
| --- | --- | --- | --- | --- | --- | --- | --- |
| UniFD | 71.04/87.14 | 58.83/61.01 | 65.20/76.79 | 54.75/54.03 | 58.35/54.87 | 55.85/56.89 | 63.87/71.97 |
| RINE | 72.00/96.53 | 58.48/88.58 | 56.95/84.50 | 50.00/60.20 | 50.00/55.10 | 50.60/59.70 | 62.20/86.20 |
| base | 95.43/99.83 | 94.85/99.20 | 85.37/94.67 | 66.05/78.10 | 64.65/80.66 | 50.39/54.56 | 88.27/94.26 |
| iCaRL | 96.00/99.65 | 91.57/97.61 | 78.11/93.34 | 96.01/99.85 | 94.99/99.80 | 94.52/99.12 | 92.40/98.26 |
| CBRS | 95.15/99.68 | 93.15/98.60 | 77.58/94.24 | 95.23/99.78 | 96.58/99.97 | **96.02/99.50** | 92.66/98.73 |
| HGR | **97.12/99.81** | **94.00/99.07** | **82.31/94.29** | **97.06/99.77** | **98.07/99.99** | 95.18/99.07 | **94.38/98.92** |
| Oracle | 95.54/99.76 | 95.62/99.39 | 78.39/96.08 | 94.57/98.86 | 98.67/99.94 | 94.90/99.08 | 93.75/99.15 |

### 4.3 CONTINUAL ADAPTATION TO 3DGS DOMAINS

We evaluate sequential adaptation to three 3DGS domains (GHA, SA, GAGAvatar), averaging over all six permutations of arrival order, and report group means for Diffusion, GANs, and Others (Deepfake, SAN), per-domain 3DGS columns, and an all-targets average over 18 datasets. In our setup, *base* trains only on SDV1.4 or ProGAN without 3DGS and *Oracle* is jointly trained on the base plus 3DGS domains in a non-continual manner. We also report the performance of the pretrained UniFD and RINE models. We can see that the pretrained UniFD and RINE models struggle to detect synthetic images generated by 3DGS. *iCaRL*, *CBRS*, and our *HGR* are sampling methods for the replay buffer whose scores are averaged across arrival orders. In the tables, boldface marks the best mACC among sampling methods only (iCaRL, CBRS, HGR). With an SDV1.4 start (Table 3), HGR achieves the highest overall mean among sampling methods and even surpasses the non-continual oracle; on 3DGS, HGR leads on GHA and SA, while CBRS is slightly higher on GAGAvatar. With a ProGAN start (Table 4), HGR again delivers the best overall mean among sampling methods and clearly improves the GANs group relative to the base; within 3DGS, CBRS tops GHA and GAGAvatar, whereas HGR peaks on SA. The base and Oracle serve as reference points rather than direct competitors; across both initializations, HGR is the most effective sampling strategy, with CBRS offering targeted gains on specific 3DGS regimes.

### 4.4 ABLATION ON HSIC COMPONENTS AND INTERMEDIATE FEATURES

Table 5 examines (left) the roles of HSIC$(x, z)$, HSIC$(y, z)$, and intermediate ViT/CLIP representations, and (right) the choice of HSIC kernel/bandwidth. Enforcing HSIC$(y, z)$ consistently strengthens cross-generator generalization by aligning the latent with task-relevant variation, while HSIC$(x,$

Table 4: **Training from ProGAN**. Cells show **mACC/mAP** (%). In our setup, *base* trains only on ProGAN without 3DGS; *Oracle* is jointly trained on ProGAN plus {GHA, SA, GAGAvatar} in a non-continual manner; and *iCaRL*, *CBRS*, and our *HGR* are sampling methods for the replay buffer. **Bold** highlights the best mACC among the sampling methods only.

| Method | Diffusion | GANs | Others | GHA | SA | GAGAvatar | Average |
|--------|-----------|------|--------|-----|-----|-----------|---------|
| UniFD | 67.03/84.34 | 95.44/99.11 | 61.95/82.11 | 53.10/56.33 | 86.10/95.41 | 65.45/76.54 | 76.14/87.64 |
| RINE | 79.21/96.58 | 97.85/99.90 | 74.42/96.42 | 54.20/66.00 | 79.30/92.90 | 61.00/77.30 | 82.50/94.69 |
| base | 93.29/97.02 | 89.48/99.99 | 88.05/97.38 | 51.25/74.75 | 55.78/94.09 | 61.98/73.83 | 85.28/95.36 |
| iCaRL | 76.79/91.10 | 88.84/94.74 | 78.90/92.80 | 95.23/99.86 | 97.47/99.98 | 96.93/99.85 | 84.33/93.96 |
| CBRS | 80.33/92.97 | 89.99/95.34 | 77.55/90.63 | **97.82/99.85** | 98.07/100.00 | **98.09/99.89** | 86.18/94.65 |
| HGR | **82.87/94.33** | **93.94/98.85** | **80.99/90.72** | 94.71/99.47 | **99.45/100.00** | 95.47/99.26 | **88.63/96.31** |
| Oracle | 82.15/95.58 | 96.10/99.72 | 85.15/97.11 | 90.90/96.66 | 98.81/99.96 | 90.98/97.06 | 89.04/98.27 |

Table 5: **HSIC ablations (components/intermediate on the left; kernel/bandwidth on the right)**. We report **mACC/mAP** (%) on **SDV1.4** and **ProGAN**. (a) toggles HSIC$(x, z)$, HSIC$(y, z)$, and the use of intermediate ViT features; the full configuration (both HSIC terms + intermediates) attains the best overall performance. (b) compares HSIC kernels and bandwidths ($\sigma$); an RBF with the median heuristic performs best and is adopted as default. **Bold** marks the top mACC per dataset.

(a) Ablations on HSIC components and intermediate features. ✓= enabled, ✗= disabled. We report **mACC/mAP** (%) on **SDV1.4** and **ProGAN**.

| HSIC$(x, z)$ | HSIC$(y, z)$ | Intermed. | SDV1.4 | ProGAN |
|--------------|--------------|-----------|--------|--------|
| ✗ | ✗ | ✗ | 89.20/96.77 | 83.30/90.26 |
| ✓ | ✗ | ✗ | 89.75/96.66 | 83.29/90.22 |
| ✗ | ✓ | ✗ | 91.64/97.34 | 87.05/94.45 |
| ✗ | ✗ | ✓ | 92.22/98.31 | 89.22/95.62 |
| ✓ | ✓ | ✗ | 91.69/97.10 | 86.66/94.10 |
| ✗ | ✓ | ✓ | 93.39/98.80 | 90.67/97.78 |
| ✓ | ✗ | ✓ | 90.27/97.48 | 87.12/93.70 |
| ✓ | ✓ | ✓ | **93.86/98.89** | **91.07/98.25** |

(b) Ablations on HSIC kernel and bandwidth. "IMQ" denotes the inverse multiquadratic kernel. We report **mACC/mAP** (%) on **SDV1.4** and **ProGAN**.

| Kernel | SDV1.4 | ProGAN |
|--------|--------|--------|
| Cosine | 88.70/98.64 | **91.49/97.22** |
| IMQ | 92.79/98.59 | 89.23/97.50 |
| RBF ($\sigma = 1$) | 92.75/98.72 | 88.82/97.89 |
| RBF ($\sigma = 2$) | 92.71/98.55 | 90.50/97.52 |
| RBF ($\sigma = 3$) | 93.14/98.71 | 90.83/97.48 |
| RBF ($\sigma = median$) | **93.86/98.89** | 91.07/98.25 |

$z$) acts as a complementary regularizer that discourages input-anchored shortcuts and stabilizes optimization; using both terms together yields a representation that preserves prior competencies yet remains adaptable. Enabling intermediate features improves both accuracy and ranking across all configurations, indicating that earlier layers expose generator cues that the HSIC bottleneck can regularize toward invariance.

For the dependence measure, Cosine and IMQ perform competitively, but an RBF kernel with a median heuristic provides the most stable behavior across both SDV1.4 and ProGAN bases, likely due to its scale adaptivity without manual tuning. For additional ablation studies, please refer to Appendix A.

## 4.5 DOMAIN-INCREMENTAL LEARNING ANALYSIS

In this section, we study the forward and backward transfer of HGR. We report per-dataset detection performance (mACC/mAP) for the arrival order whose final mACC is closest to the mean over all six permutations of arrival orders; in each subtable, the first row lists the chosen sequence (starting from the base), and the column shows the performance after the dataset has arrived. Forgetting concentrates on out-of-domain targets; for example, when training from SDV1.4, degradation is most pronounced on GANs and Others (and symmetrically, diffusion degrades when starting from Pro-GAN), though the effect remains modest under our rehearsal budget. At the same time, new domains provide positive transfer by exposing the model to additional artifact patterns and scene statistics, yielding consistent gains on datasets related to the most recently trained domain and smaller but nontrivial improvements on some cross-generator targets.

Table 6: **Domain-incremental learning analysis**. Each subtable shows the arrival order whose final mACC is closest to the mean over all six permutations of arrival orders; the first row lists the selected sequence, and the column shows the performance after the dataset has arrived. The numbers highlighted in gray indicate that their corresponding 3DGS datasets have not yet been included in the domain-incremental training.

(a) Starting from SDV1.4, then SA, GAGAvatar, and GHA arrived sequentially. Forgetting happened mostly on GANs and Others.

| Dataset | SDV1.4 | SA | GAGAvatar | GHA |
|---|---|---|---|---|
| Diffusion | 95.43/99.83 | 97.41/99.84 | 97.59/99.81 | 96.90/99.79 |
| GANs | 94.85/99.20 | 93.74/98.60 | 94.36/99.12 | 93.24/98.79 |
| Others | 85.37/94.67 | 81.01/96.35 | 76.79/93.93 | 84.43/93.06 |
| SA | 64.65/80.66 | 97.01/99.83 | 99.86/100.00 | 98.99/100.00 |
| GAGAvatar | 50.39/54.56 | 64.91/69.93 | 99.37/99.98 | 94.17/99.51 |
| GHA | 66.05/78.10 | 66.67/86.96 | 66.41/84.82 | 98.80/99.94 |
| Average | 88.27/94.26 | 90.83/96.66 | 92.70/98.11 | 94.36/98.71 |

(b) Starting from ProGAN, then SA, GHA, and GAGAvatar arrived sequentially. Forgetting happened mostly on Diffusion and Others.

| Dataset | ProGAN | SA | GHA | GAGAvatar |
|---|---|---|---|---|
| GANs | 89.48/99.99 | 91.96/99.77 | 89.56/99.42 | 93.18/99.39 |
| Others | 88.05/97.38 | 76.84/94.96 | 83.28/94.53 | 82.82/92.31 |
| Diffusion | 93.29/97.02 | 85.33/94.85 | 87.67/94.75 | 84.66/93.63 |
| SA | 55.78/94.09 | 94.17/100.00 | 100.00/100.00 | 99.55/100.00 |
| GHA | 51.25/74.75 | 66.71/88.29 | 96.23/99.93 | 92.57/99.03 |
| GAGAvatar | 61.98/73.83 | 50.23/71.24 | 55.18/68.98 | 98.49/99.88 |
| Average | 85.28/95.36 | 84.10/95.11 | 87.17/95.43 | 89.33/96.40 |

## 5 DISCUSSION

### 5.1 FUTURE WORK

In this section, we discuss several special scenarios that we have not yet explored. First, we do not evaluate extremely low-resolution inputs (e.g., $32 \times 32$, $64 \times 64$) or heavily post-processed images (e.g., strong JPEG compression, Gaussian blur). Our detector is built on CLIP ViT, which assumes inputs of at least $224 \times 224$, and the synthetic image datasets we use typically have resolutions between $256 \times 256$ and $512 \times 512$. In practice, we envision handling such cases with a separate robustness module tailored to severe degradations. Second, we do not evaluate adversarial robustness (e.g., FGSM/PGD attacks) in this work. A natural extension is to combine HSIC-Guided Replay with adversarial training: adversarially perturbed real and synthetic images could be injected into the replay buffer. Third, our 3DGS experiments currently assume balanced real/synthetic splits. Studying highly imbalanced regimes (e.g., 1:5, 1:10) and comparing different rebalancing strategies (class weighting, resampling, or hybrid schemes) is left for future work.

### 5.2 EXTENSIONS

This paper focuses on synthetic image detection. Extending the HSIC bottleneck to multi-modal inputs, such as synthetic 3D point clouds, is an exciting direction for future work. At present, we aggregate CLIP intermediate and final layers by simple concatenation; exploring alternative aggregation schemes (e.g., attention-based fusion) is another promising avenue. We focus on CLIP ViT backbones in this work and do not evaluate lighter models such as MobileNetV3 or EfficientNet-B0. Since the HSIC bottleneck itself is backbone-agnostic, applying it to lightweight architectures for edge deployment is a natural extension. Finally, we do not evaluate on real-world platforms such as Reddit. Extending our benchmarks to such noisy, in-the-wild data is important for practical deployment, and we plan to investigate this direction in future work.

## 6 CONCLUSION

We presented a detector that couples an HSIC bottleneck on intermediate CLIP features with HSIC-Guided Replay regularized by a $k$-center coverage term. The bottleneck filters text-alignment nuisances while preserving discriminative cues, yielding strong cross-generator generalization when training on either diffusion or GAN sources. The replay mechanism selects compact, informative exemplars that stabilize continual adaptation to 3D Gaussian Splatting (3DGS) domains. Across SDV1.4- and ProGAN-based training, our method achieves consistent gains in both mACC and mAP, maintains performance on earlier targets, and shows low variance under different 3DGS orderings. To support rigorous study of rendered-fakes, we curated three 3DGS head avatar datasets spanning multi-view reconstruction, single-view reconstruction, and a generative method. To the best of our knowledge, we are the first to introduce a photorealistic 3DGS Synthetic Image Benchmark, together with a continual learning solution tailored to this setting.

## ACKNOWLEDGEMENTS

This work was supported in part by NSTC Graduate Research Fellowship, NSTC grants 112-2221-E-A49-100-MY3, 113-2221-E-001-010-MY3, 113-2221-E-002-112-MY3 of Taiwan, NTU grant NTU-CC 115L890903 and Taiwan Centers of Excellence (TCE). We thank the National Center for High-performance Computing for providing computing resources.

## ETHICS STATEMENT

Our study uses only publicly available image datasets containing human faces, accessed and used under the research licenses of the datasets. We conducted no interaction or intervention with individuals and accessed no private or non-public data. We process facial imagery solely for research on synthetic image detection, apply data minimization and security safeguards consistent with the dataset licenses, and report only aggregate results. We do not attempt to identify, profile, or target individuals. We have carefully considered potential impacts and do not anticipate ethical risks beyond those commonly encountered in computer vision and machine learning research.

## REPRODUCIBILITY STATEMENT

We document the training pipeline and evaluation protocols in both the main paper and the appendix, with explicit hyperparameters and dataset partitions to support exact replication. We also describe implementation settings and reporting conventions to match our results. To further enable reproducibility, we will release the full training and evaluation code, along with runnable scripts.

## THE USE OF LARGE LANGUAGE MODELS (LLMS)

We used LLMs to assist with (i) polishing prose (grammar, flow, and clarity), (ii) reorganizing and tightening section structure, captions, and titles, and (iii) brainstorming keywords and query strings to surface related work.

For literature discovery, the model suggested search terms and produced brief summaries to orient the authors. Every citation in the paper was located through standard search engines or digital libraries and then read and verified by the authors; we did not accept model-generated references without inspection. Numerical results, comparisons, and quotes were cross-checked against the original sources.

We edited all model-suggested text for accuracy and originality and ensured that the writing reflects our intent. No confidential or sensitive data were shared with the model. The final manuscript, including all tables and figures, was reviewed end-to-end by the authors for factual correctness and completeness.

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

# A  APPENDIX

## A.1  ROBUSTNESS OF CROSS-GENERATOR SYNTHETIC IMAGE DETECTION

Figure 3 plots mean accuracy with across-seed standard deviation shown as error bars. Variation is consistently small: for most targets, the standard deviation is roughly 0 to 1.5 percentage points, with slightly larger yet still bounded spreads on a few harder sets such as Midjourney, GauGAN, and Deepfake. Overall accuracies remain high, typically in the 80 to 90% range, and several targets approach 100%. The tight uncertainty bands indicate robustness to random initialization and optimization noise and support the conclusion that our HSIC-based training yields reliable cross-generator synthetic image detection.

## A.2  PRETRAINED MODEL CHOICE

Figure 4 compares three feature extractors: CLIP ViT-L/14, CLIP ViT-B/16, and DINOv2 ViT-L/14. CLIP ViT-L/14 is our default and consistently delivers the highest accuracy across most datasets. CLIP ViT-B/16 follows closely, typically trailing by about 1 to 3 percentage points, which indicates that the method is not overly sensitive to model scale within the CLIP family. DINOv2 ViT-L/14 lags more clearly, particularly on targets produced by GANs and on Deepfake, where the gap can widen to roughly 10 to 30 percentage points. These trends suggest that image-text pretraining in CLIP exposes generator-related artifacts more effectively, while the improvements we observe are largely attributable to the HSIC-based training objective rather than reliance on a single backbone.

## A.3  BLOCK-WISE EFFECTS OF INTERMEDIATE FEATURES

Figure 5 reports performance when evaluation is restricted to a single CLIP ViT block. At inference time, we zero out features from all non-target blocks and retain only the target block.

Across most GAN families (ProGAN, StyleGAN, StarGAN; and to a lesser extent CycleGAN and BigGAN), we observe a broad mid-to-late plateau: many adjacent blocks yield comparable mean accuracy, indicating robustness to modest shifts in the chosen block. In contrast, GauGAN exhibits a narrow late-layer peak with steep degradation outside that region. Because our protocol fixes the intermediate block globally for efficiency and fairness, the choice that best serves the majority becomes suboptimal for GauGAN—accounting for its weaker row in Table 2.

When GauGAN performance is prioritized, two lightweight adjustments help: (i) aggregate a small window of adjacent late blocks to form a multi-block feature; or (ii) apply an HSIC pyramid that regularizes over neighboring blocks, capturing late-layer signal while preserving the plateaued behavior on other models.

## A.4  ANALYSIS OF LEARNED REPRESENTATIONS

Figure 6 and Figure 7 visualize feature geometry before and after applying the HSIC bottleneck using t-SNE. For each target dataset, the upper panel shows pretrained CLIP embeddings x and the lower panel shows learned embeddings z after training on SDV1.4 in Figure 6 and on ProGAN in Figure 7; blue denotes real and orange denotes synthetic. Relative to x, z forms tighter within-class clusters and larger margins between real and synthetic across generators, with the effect most visible for transfer from diffusion to GAN and from GAN to diffusion. This matches the objective: the HSIC term with labels increases between-class separation, while the HSIC term with inputs reduces reliance on input-specific artifacts and lowers within-class variance. When trained on SDV1.4, real clusters in z become more generator-agnostic, and synthetic clusters move closer together, indicating stronger alignment across models. When trained on ProGAN, z sharpens separation on GAN targets and increases margins on diffusion targets, though small pockets of overlap remain for certain datasets, such as GauGAN. Overall, these t-SNE views corroborate the quantitative results by showing that the HSIC bottleneck reshapes features toward a generator-invariant yet discriminative structure.

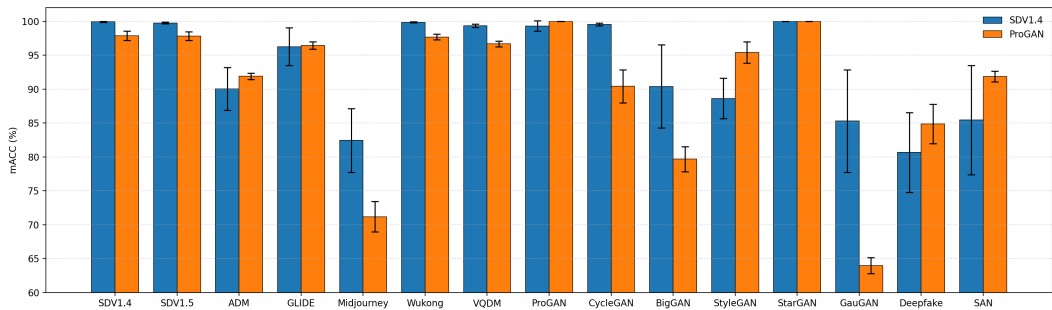

Figure 3: **Per-dataset accuracy with variability**. Bars show the mean accuracy over 5 runs; error bars denote the across-seed standard deviation. Colors compare detectors trained on **SDV1.4** (blue) versus **ProGAN** (orange).

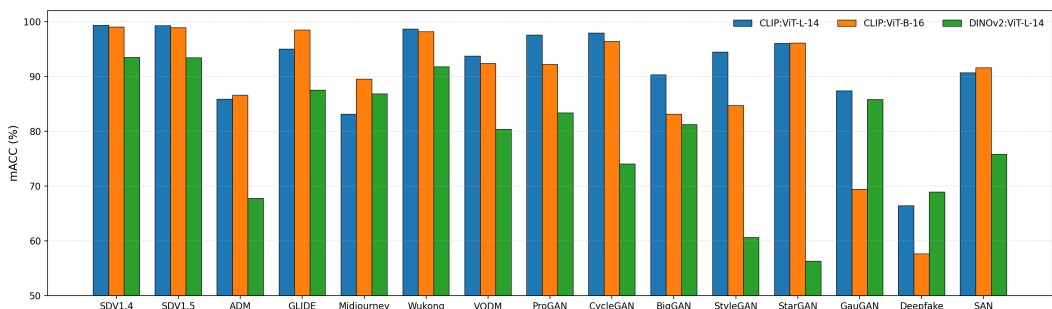

Figure 4: **Pretrained model choice**. We extract frozen features from three pretrained models—CLIP ViT-L/14, CLIP ViT-B/16, and DINOv2 ViT-L/14—and train the detector on SDV1.4. The figure reports accuracy for each target dataset.

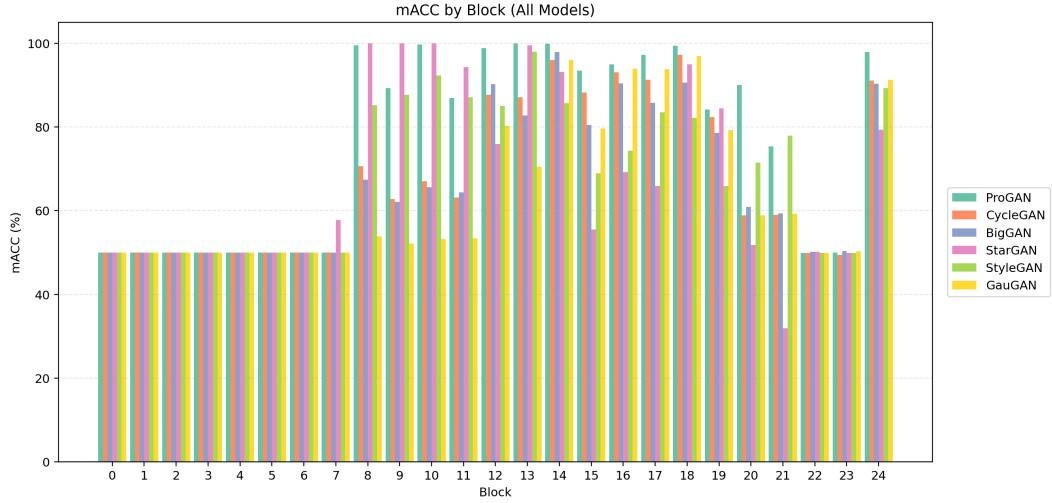

Figure 5: **Where the signal lives: block-wise test accuracy across GAN families**. Each group reports accuracy when the detector uses only one CLIP ViT intermediate block (non-target blocks are zeroed at inference).

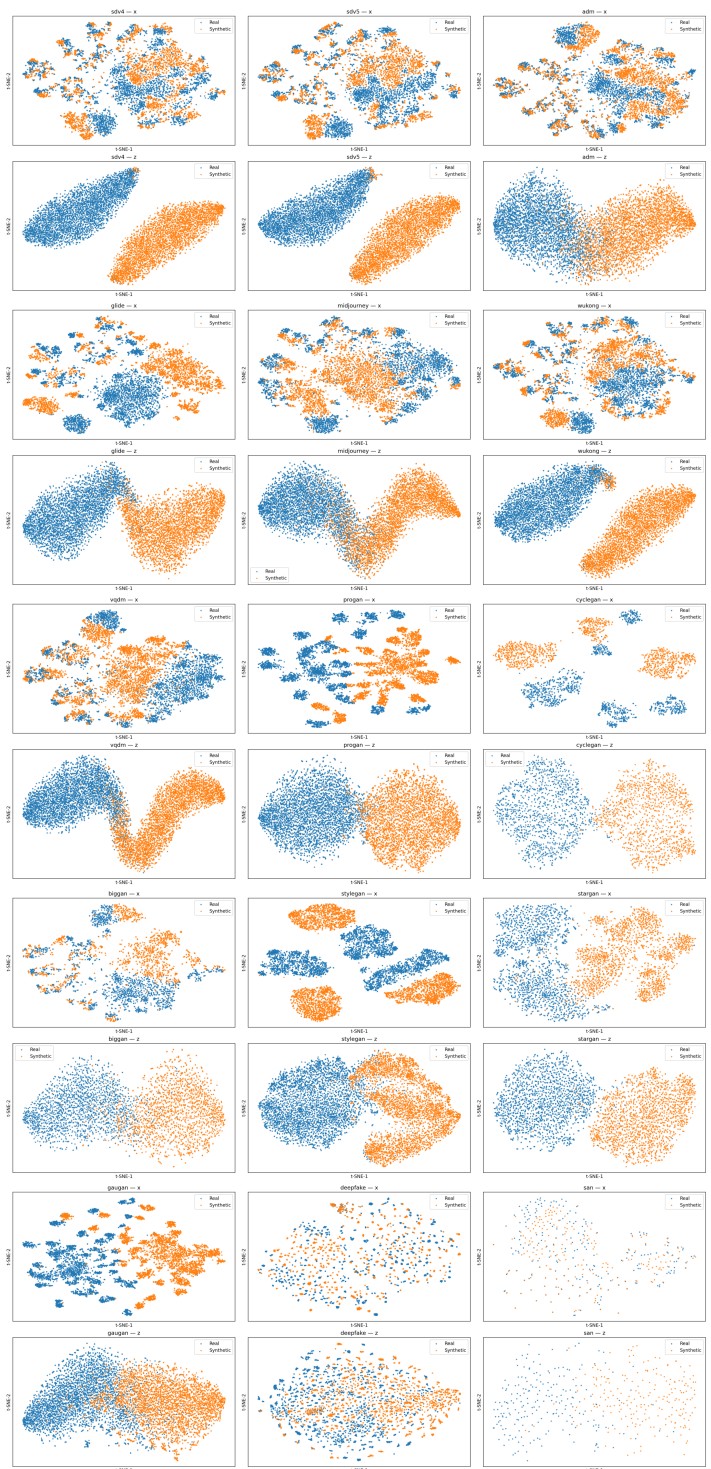

Figure 6: **t-SNE of test features across generators (upper: $x$, lower: $z$), trained on SDV1.4**. For each test set (columns grouped by generator; rows per dataset), the upper panel shows embeddings of the pretrained CLIP features $x$, while the lower panel shows embeddings of the features $z$ learned by training on SDV1.4 with the HSIC bottleneck. Points are colored by ground truth (blue = real, orange = synthetic). Relative to $x$, the HSIC-trained $z$ generally yields tighter, better-separated clusters of real versus synthetic across both diffusion and GAN generators, indicating improved transferable separability.

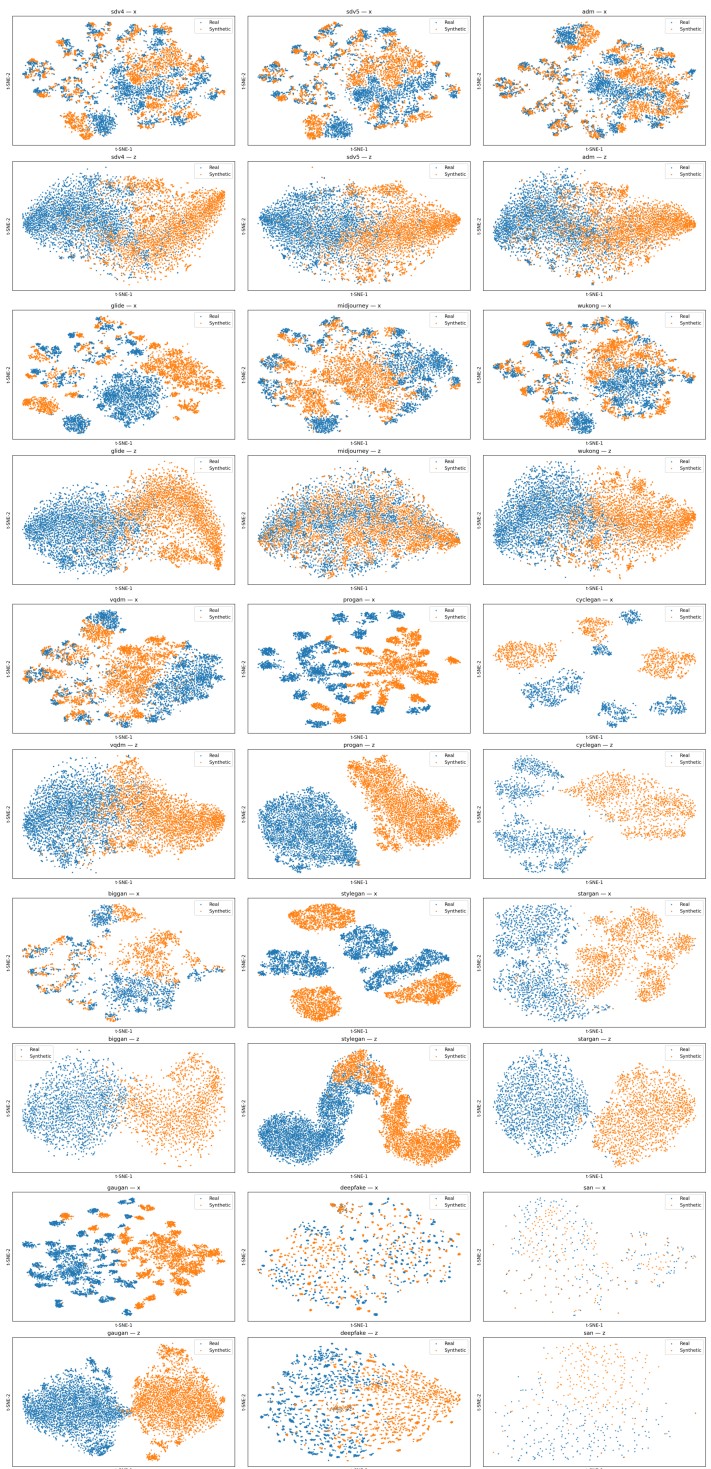

Figure 7: **t-SNE of test features across generators (upper: $x$, lower: $z$), trained on ProGAN**. For each test set, the upper panel visualizes embeddings of the pretrained CLIP features $x$, and the lower panel shows embeddings of the features $z$ learned by training with the HSIC bottleneck on **ProGAN**. Points are colored by ground truth (blue = real, orange = synthetic). Relative to $x$, the HSIC-trained $z$ typically exhibits tighter clusters and larger margins between real and synthetic across both GAN and diffusion generators, indicating improved transferable separability from a ProGAN-trained detector.

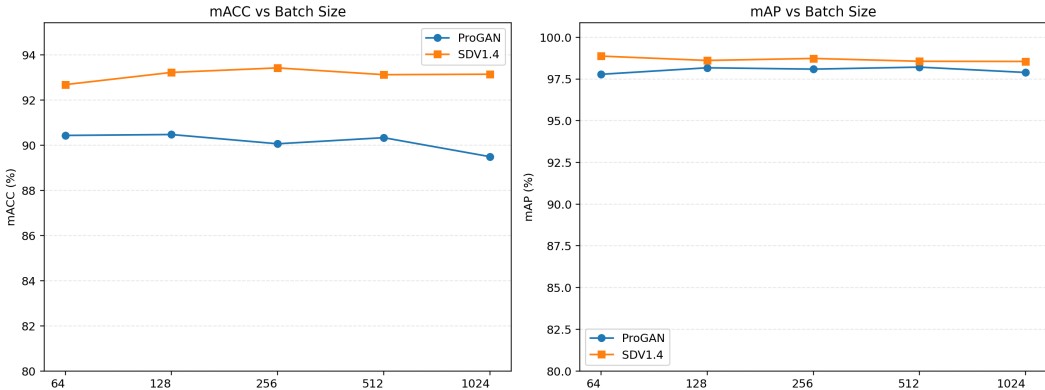

Figure 8: **Batch-size ablation for the HSIC bottleneck**. Mini-batch size varies across $\{64, 128, 256, 512, 1024\}$ with all other settings fixed. The left panel reports mean accuracy and the right panel reports mean average precision. Curves are shown for SDV1.4 in orange and ProGAN in blue. Moderate batches deliver the strongest mean accuracy—around 256 for SDV1.4 and between 128 and 512 for ProGAN—while very large batches reduce accuracy for the GAN start. Mean average precision remains near saturated and largely insensitive to batch size, indicating robust ranking.

## A.5   ABLATION ON BATCH SIZE

Figure 8 analyzes how mini-batch size affects cross-generator performance under the HSIC bottleneck. Two consistent patterns emerge. First, mean accuracy follows a non-monotonic trend: moving from very small to moderate batches increases accuracy, while pushing to very large batches lowers accuracy or plateau. This reflects a balance between helpful gradient noise at moderate sizes and reduced variance of the HSIC estimator at large sizes that can over-stabilize updates and diminish useful stochasticity. Second, mean average precision stays high and nearly flat across the full range, indicating that ranking is robust even when the classification boundary shifts with batch size.

We adopt a moderate batch size of 128 by default to balance generalization and the computational cost of the quadratic HSIC term. Larger batches can be used when memory allows and when prioritizing stability in average precision, but they offer limited gains in this setting and can slightly reduce accuracy.

## A.6   ABLATION ON ACTIVATION FUNCTIONS

Figure 9 examines the nonlinearity used in the HSIC bottleneck head. Mean accuracy depends on the training paradigm: when trained on ProGAN, smoother bounded activations such as sigmoid and tanh yield the most reliable accuracy, whereas when trained on SDV1.4, a linear head is preferable and ReLU tends to underperform. This pattern is consistent with the HSIC objective: ReLU induces hard sparsification that concentrates probability mass and increases the variance of the dependence estimator, while bounded nonlinearities confine the latent to a finite range that aligns better with the radial basis function HSIC kernel; a linear head preserves intermediate CLIP ViT features that already carry generator cues under diffusion training. Mean average precision remains high and changes little across activations, indicating that confidence ranking is robust even when the decision boundary shifts.

## A.7   ABLATION ON HIDDEN DIMENSION

Figure 10 examines how the width of the HSIC bottleneck affects cross-generator performance. Accuracy rises as the latent becomes wider from very small to moderate sizes, then shows diminishing returns; when trained on SDV1.4 the model peaks at a mid sized latent, while when trained on ProGAN a slightly larger latent is preferred. This reflects a balance between capacity and regularization: a small latent space limits discrimination and increases variance in the HSIC estimator, whereas a very large latent weakens the pressure toward generator invariance and adds little utility. Mean av-

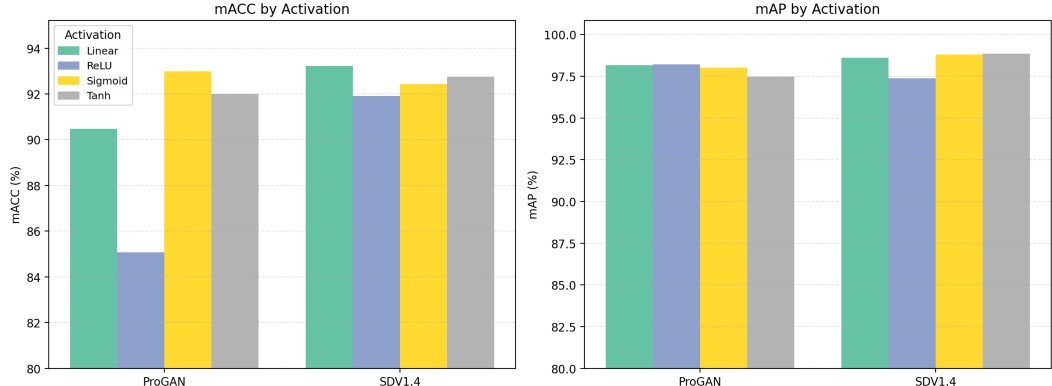

Figure 9: **Activation-function ablation for the HSIC bottleneck**. The bottleneck head uses one of four activations: Linear, ReLU, Sigmoid, or Tanh, with all other settings fixed. The left panel reports mean accuracy and the right panel reports mean average precision when trained on ProGAN or SDV1.4. When trained on ProGAN, smoother bounded activations such as Sigmoid and Tanh deliver stronger and more stable accuracy; when trained on SDV1.4, a Linear head is most effective and ReLU tends to underperform. Across both cases, mean average precision remains high and largely insensitive to the activation choice.

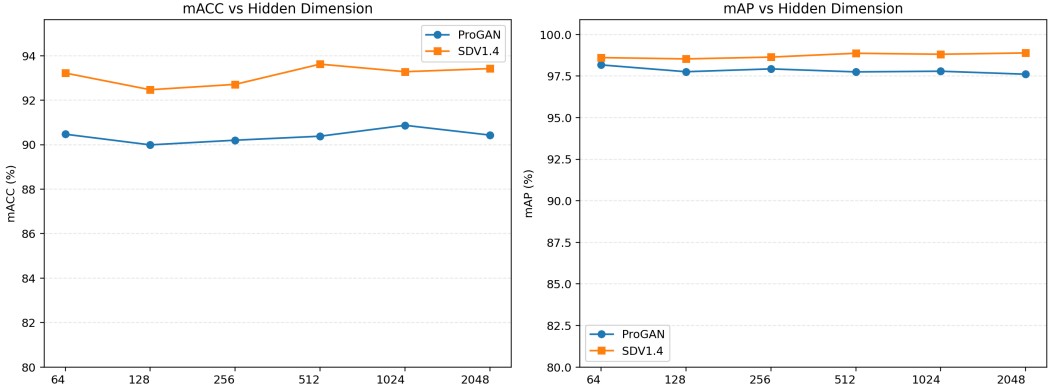

Figure 10: **Hidden-dimension ablation for the HSIC bottleneck**. The latent width of the bottleneck head varies over $\dim(z) \in \{64, 128, 256, 512, 1024, 2048\}$ with all other settings fixed, using a radial basis function HSIC with a median bandwidth and the same training schedule. The left panel reports mean accuracy and the right panel reports mean average precision; curves are shown in orange when trained on SDV1.4 and in blue when trained on ProGAN. Moderate widths perform best—when trained on SDV1.4 the model favors a mid-sized latent, and when trained on ProGAN a slightly larger latent is preferred—while very small widths underfit and very large widths offer diminishing returns. Mean average precision remains high and largely flat across widths, indicating robust ranking.

erage precision remains high and nearly flat across widths, indicating that the confidence ranking is stable even as the decision boundary shifts. Because the differences across reasonable widths are subtle, we adopt a default latent dimension of 64 to reduce parameters and runtime while preserving accuracy, and only increase the width when resources permit or a target shows a clear benefit.

## A.8    ABLATION ON NUMBER OF LAYERS

Figure 11 analyzes how the depth of the HSIC bottleneck head influences cross-generator performance. Mean accuracy improves as we move from a single linear layer to a moderate depth and then either stabilizes or declines at excessive depth; when trained on SDV1.4 the results are broadly stable once the head is moderately deep, while when trained on ProGAN performance peaks at mid depth

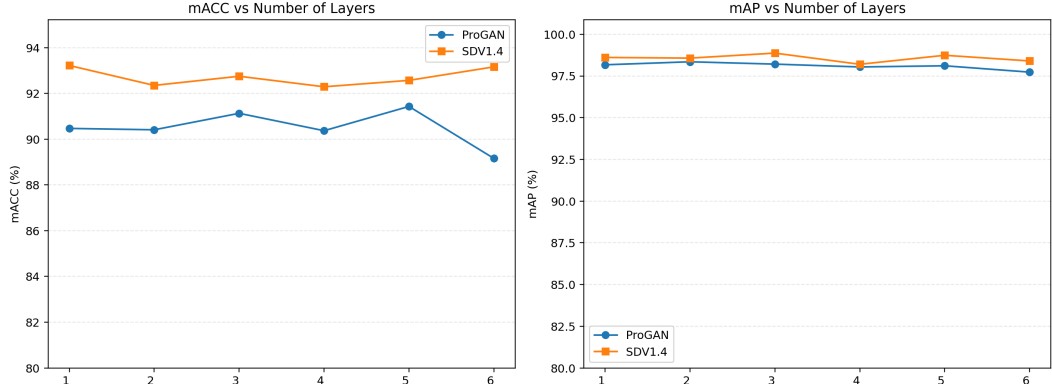

Figure 11: **Depth ablation for the HSIC bottleneck head with intermediate features**. The bottleneck depth varies over $L \in \{1, 2, 3, 4, 5, 6\}$ while width, kernel, and schedule are fixed. The left panel shows mean accuracy and the right panel shows mean average precision; curves in blue indicate training on ProGAN and curves in orange indicate training on SDV1.4. Mean accuracy is highest at moderate depth: accuracy rises from shallow heads, then plateaus for SDV1.4 and declines for very deep heads when trained on ProGAN. Mean average precision remains high and largely insensitive to depth, indicating robust ranking.

Table 7: Ablation of HSIC relevance in HSIC-Guided Replay (HGR) on the SDV1.4 source. Numbers denote (mACC/mAP).

| Method | Diffusion | GANs | Others | GHA | SA | GAGAvatar | Average |
|---|---|---|---|---|---|---|---|
| w/o HSIC relevance | 95.39/99.80 | **94.15/99.13** | **82.99/94.42** | 94.75/99.62 | **99.06/99.98** | **95.92/99.13** | 93.80/98.94 |
| HGR | **97.12/99.81** | 94.00/99.07 | 82.31/94.29 | **97.06/99.77** | 98.07/99.99 | 95.18/99.07 | **94.38/98.92** |

and drops for very deep heads. This follows the bias-variance behavior of dependence estimation: shallow heads underfit and limit the expressiveness of the latent representation, whereas very deep heads over-shape features, increase estimator variance, and erode generator invariance. Mean average precision remains high and largely unchanged across depths, showing that confidence ranking is robust even as decision boundaries shift with capacity. Because differences across reasonable depths are subtle, we use a single layer by default for both SDV1.4 and ProGAN to reduce parameters and computation, and increase depth only when clear gains are observed.

## A.9 Ablation of HSIC Relevance in HSIC-Guided Replay

We further conduct an ablation to isolate the contribution of the HSIC relevance term in equation 10. To this end, we construct a *k-center only* variant by removing the HSIC component and setting $\lambda_{\mathrm{kc}}=1$, so that exemplar selection is governed solely by the normalized coverage score $d_i(t)$, i.e., $s_i(t) = 1 - \mathcal{N}\big(d_i(t)\big)$. This configuration corresponds to a pure coreset-style selection in feature space and serves as a competitive baseline to assess the added value of HSIC-guided relevance. As shown in Tables 7 and 8, HGR improves the average performance, particularly on Diffusion and GHA domains, while the $k$-center only variant can occasionally perform slightly better on individual generators or categories.

## A.10 Effect of HSIC Bottleneck on Text-Alignment Semantics

To quantify how the HSIC bottleneck reshapes CLIP features with respect to text alignment, we measure the cosine similarity between image features and text features on the ProGAN testing set. For each class label $c \in \{\texttt{airplane}, \texttt{bicycle}, \dots, \texttt{tvmonitor}\}$, we compute the cosine similarity between the CLIP image embedding and the CLIP text embedding of the prompt "an image of $\langle c \rangle$". The blue bars in Fig. 12 correspond to the original CLIP image features (before applying the HSIC bottleneck). Across all classes, the cosine similarities are consistently around 0.18–0.24, indi-

Table 8: Ablation of HSIC relevance in HSIC-Guided Replay (HGR) on the ProGAN source. Numbers denote (mACC/mAP).

| Method | Diffusion | GANs | Others | GHA | SA | GAGAvatar | Average |
|---|---|---|---|---|---|---|---|
| w/o HSIC relevance | 77.54/93.55 | **94.35/98.89** | 77.95/89.48 | 91.49/99.36 | **99.60/99.99** | **96.31/99.53** | 86.23/95.89 |
| HGR | **82.87/94.33** | 93.94/98.85 | **80.99/90.72** | **94.71/99.47** | 99.45/100.00 | 95.47/99.26 | **88.63/96.31** |

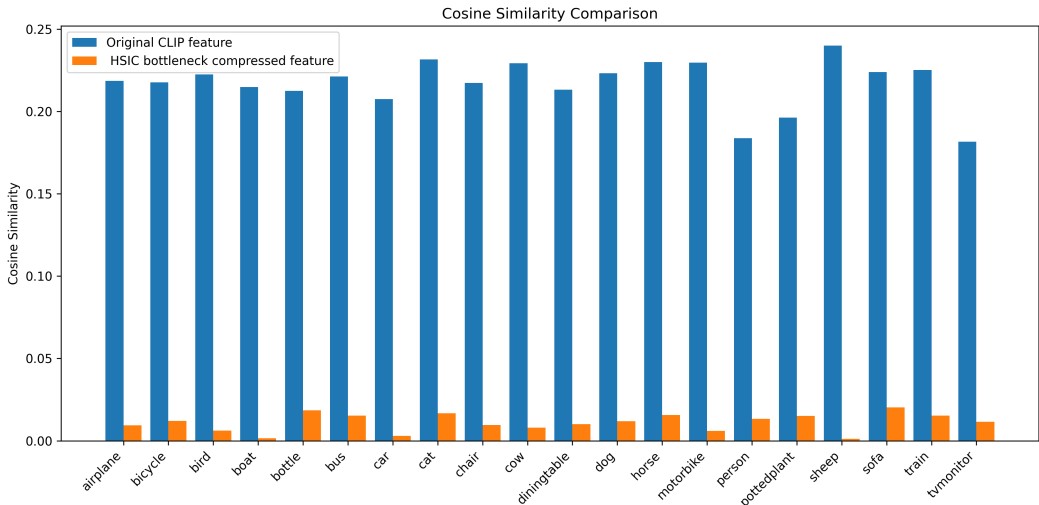

Figure 12: Cosine similarity between image and text features on the ProGAN test set. For each class $c$, we compute the cosine similarity between the CLIP image feature and the CLIP text feature of the prompt "an image of $\langle c \rangle$". **Blue:** original CLIP features, which exhibit strong text alignment across all classes. **Orange:** HSIC bottleneck compressed features, whose cosine similarity to the same text prompts is dramatically reduced, showing that the HSIC bottleneck suppresses text-alignment semantics.

cating that the original features preserve strong text-alignment semantics even for ProGAN synthetic images.

After training with the HSIC bottleneck loss, we recompute cosine similarities using the compressed features (orange bars in Fig. 12). These similarities drop to around 0.01 for every class, approaching zero. This substantial and systematic reduction demonstrates that the HSIC bottleneck effectively suppresses the alignment between image features and the corresponding text prompts.

## A.11 SENSITIVITY OF HSIC BOTTLENECK WEIGHTS

Figs. 13 and 14 visualize a two-dimensional grid search over the HSIC bottleneck weights $\lambda_x$ and $\lambda_y$ on SDv1.4 and ProGAN, respectively. For each $\lambda_x, \lambda_y \in \{100, 200, \ldots, 1000\}$, we train a detector with the HSIC bottleneck and report the resulting mean accuracy (mACC) and mean average precision (mAP).

On SDv1.4, both mACC and mAP remain consistently high across a broad range of weights, with mACC fluctuating within roughly one percentage point and mAP remaining close to $99\%$. The best-performing configurations tend to appear in the mid-to-high range of $\lambda_x$ and $\lambda_y$, but the overall heatmap exhibits a wide plateau rather than sharp peaks. This indicates that the detector is relatively insensitive to the exact choice of HSIC weights and that the bottleneck provides stable gains over a large region of the hyperparameter space.

A similar trend is observed on ProGAN. The mACC heatmap shows only minor variation (on the order of 0.5–1.0 percentage point) across the entire grid, while mAP remains around $98\%$ for most settings, with slightly lower values only at a few extreme combinations. Again, the central region

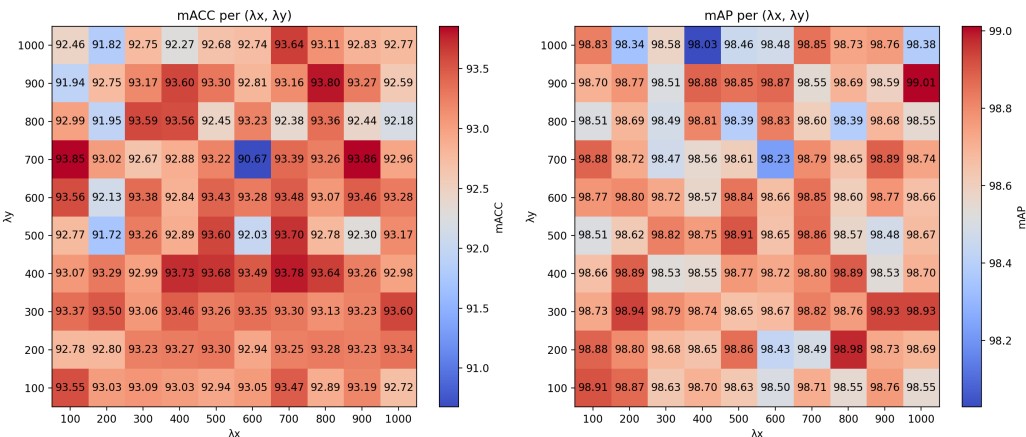

Figure 13: Grid search over the HSIC bottleneck weights $\lambda_x$ and $\lambda_y$ on SDv1.4. We sweep $\lambda_x, \lambda_y \in 100, 200, \dots, 1000$ and report the resulting mean accuracy (mACC, left) and mean average precision (mAP, right) of the detector trained on SDv1.4-generated images.

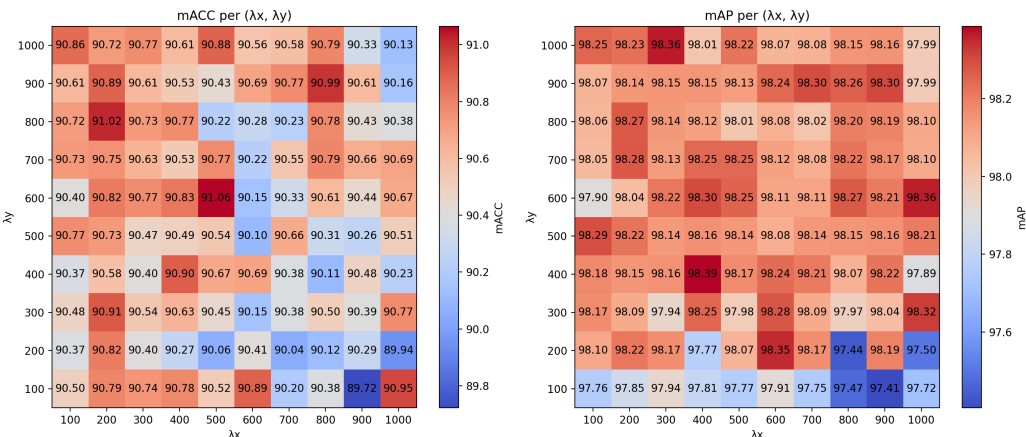

Figure 14: Grid search over the HSIC bottleneck weights $\lambda_x$ and $\lambda_y$ on ProGAN. We sweep $\lambda_x, \lambda_y \in 100, 200, \dots, 1000$ and report the resulting mean accuracy (mACC, left) and mean average precision (mAP, right) of the detector trained on ProGAN-generated images.

of the grid yields near-optimal performance, suggesting that the method does not rely on fine-tuning $\lambda_x$ and $\lambda_y$ to a narrow sweet spot.

Overall, these experiments demonstrate that the HSIC bottleneck is robust to a wide range of weight choices on both SDv1.4 and ProGAN. In the remainder of the paper, we therefore fix $\lambda_x$ and $\lambda_y$ to a single representative configuration from the high-performing plateau for all experiments.

## A.12 CLUSTERING STRUCTURE OF HSIC BOTTLENECK REPRESENTATIONS

To quantitatively assess how the HSIC bottleneck organizes different generator families in feature space, we compute two clustering metrics on the embeddings: the silhouette coefficient (higher is better, range $[-1, 1]$) and the Davies–Bouldin (DB) index (lower is better). We group generators into three families—Diffusion, GANs, and Others (deepfake and SAN)—and report the mean silhouette / DB score for each family in Table 9. The first row ($x$) uses the original CLIP image features, while the second and third rows ("SDV1.4 $z$" and "ProGAN $z$") use the HSIC bottleneck representations trained on SDV1.4 and ProGAN, respectively.

Table 9: Silhouette coefficient (higher is better) and Davies–Bouldin index (lower is better) computed image embeddings, grouped by generator family (Diffusion, GANs, Others). The first row ($x$) uses the original CLIP image features, while the second and third rows ("SDV1.4 $z$" and "ProGAN $z$") use HSIC bottleneck representations trained on SDV1.4 and ProGAN, respectively. Best values in each column are bolded.

| Embedding | Training Source | Diffusion | GANs | Others |
|:---:|:---:|:---:|:---:|:---:|
| $x$ | - | 0.03 / 5.64 | 0.08 / 3.75 | 0.04 / 5.37 |
| $z$ | SDV1.4 | **0.65 / 0.45** | 0.59 / 0.53 | 0.30 / 1.40 |
| $z$ | ProGAN | 0.34 / 1.19 | **0.69 / 0.40** | **0.34 / 0.98** |

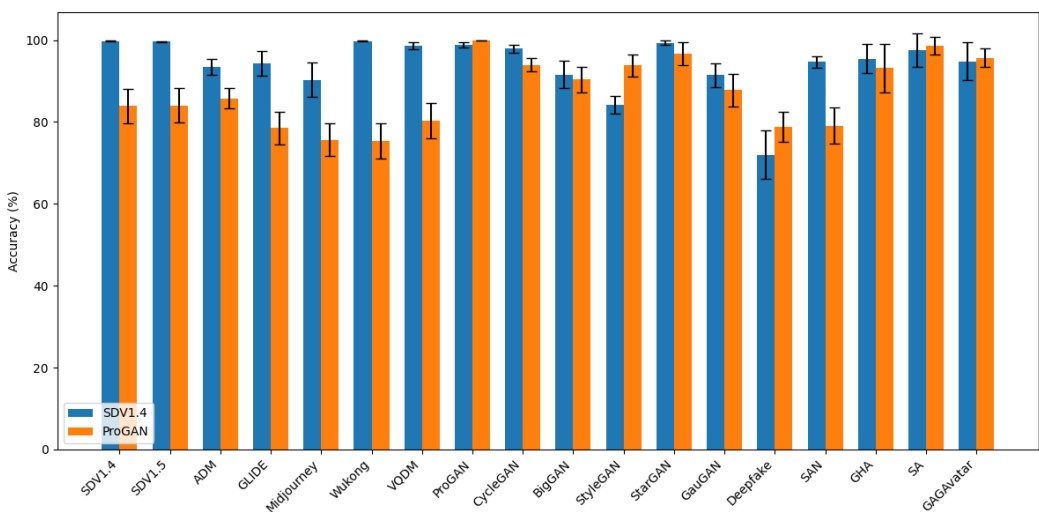

Figure 15: **Robustness of HSIC-Guided Replay (HGR).** Bars show the mean accuracy (%) over 5 independent runs of HGR for each target generator, when trained with SDV1.4 (blue) or ProGAN (orange) as the source domain. Error bars denote the standard deviation across runs.

The raw CLIP features ($x$) exhibit very weak clustering structure across all families, with near-zero silhouette scores (0.03–0.08) and large DB indices (3.75–5.64), indicating that real and synthetic images are heavily entangled in the original CLIP embedding space. In contrast, both HSIC bottleneck variants substantially improve the clustering quality: silhouette scores increase to 0.30–0.69 and DB indices drop below 1.5 for all groups. Moreover, the source domain used to train the bottleneck leaves a clear imprint on the structure: the SDV1.4-trained bottleneck yields the best clustering for diffusion models (silhouette 0.65, DB 0.45), whereas the ProGAN-trained bottleneck achieves the best scores on GANs and Others. These results show that the HSIC bottleneck not only suppresses text-alignment semantics, but also induces a more generator-aware structure in the representation space that reflects the training source.

## A.13 ROBUSTNESS OF HSIC-GUIDED REPLAY

Figure 15 summarizes the results for HGR trained with SDV1.4 or ProGAN as the source domain. Across a diverse set of diffusion models, GANs, deepfake generators, and 3DGS-based head avatar models (GHA, SA, GAGAvatar), HGR achieves consistently high accuracy with small variance across runs. This indicates that our improvements are not due to a favorable random seed, and that HGR yields stable performance under repeated training.

## A.14 IMPLEMENTATION DETAILS

Models are implemented in PyTorch and trained on a single NVIDIA GPU with SGD (learning rate $10^{-4}$). The encoder $f_{\theta_f}$ is a single fully connected (linear) layer with input dimension 19200 and output dimension 64, and the classifier $g_{\theta_g}$ is also a single fully connected (linear) layer with input dimension 64 and output dimension 1; no activation functions are used. The HSIC bottleneck uses Gaussian RBF kernels with bandwidth set by the median heuristic. We set $\lambda_x = 900$ and $\lambda_y = 700$ when training on SDV1.4 and $\lambda_x = 500$ and $\lambda_y = 600$ when training on ProGAN. We pick $m_c$ exemplars per class $c$ with $m_0 + m_1 = \lceil \text{keep\_frac} \cdot N \rceil$, where $N$ is the total number of training samples and we set keep_frac=0.01. We first enforce class balance and then assign any remainder to the larger class.

## A.15 GENERATIVE MODELS

Our evaluation covers a broad spectrum of generative models spanning early GAN-based architectures, semantic image-to-image models, and recent large-scale diffusion and text-to-image systems. Below, we briefly summarize the key characteristics of each family.

**GAN-based image generators.** **ProGAN** (Karras et al., 2018) introduces a curriculum-style training schedule where both generator and discriminator are grown from low to high resolutions, greatly stabilizing GAN training and enabling $1024^2$-resolution image synthesis on datasets such as CelebA-HQ. **StyleGAN** (Karras et al., 2019) builds on ProGAN with a style-based generator architecture that maps a latent vector into an intermediate space and modulates features via AdaIN at each layer, yielding highly controllable and disentangled factors of variation for high-resolution face generation. **BigGAN** (Brock et al., 2019) scales class-conditional GAN training to large ImageNet-scale datasets, combining architectural and regularization tricks (e.g., orthogonal regularization, truncation) to significantly improve fidelity and diversity in high-resolution synthesis. **StarGAN** (Choi et al., 2018) and **CycleGAN** (Zhu et al., 2017) focus on image-to-image translation. CycleGAN learns mappings between two unpaired domains using cycle-consistency losses. StarGAN generalizes this idea to multi-domain translation with a single generator–discriminator pair, controlled by a domain label input. **GauGAN** (Park et al., 2019) takes structured inputs such as semantic segmentation maps and uses spatially-adaptive normalization layers to preserve semantic information throughout the network, producing photorealistic scene-level images from coarse layouts.

**Deepfake and super-resolution generators.** Beyond generic image synthesis, we also consider models that manipulate or enhance existing images. **Deepfake** facial manipulation is represented in our setup via methods from **FaceForensics++** (Rossler et al., 2019), which encompasses several face-editing pipelines (e.g., reenactment, replacement, and neural texture–based manipulation) and has become a de facto benchmark for evaluating forgery detectors on human faces. Although **SAN** (Dai et al., 2019) is not a generative model in the strict sense, it hallucinates high-frequency details from low-resolution inputs through second-order attention mechanisms. As such, SAN can act as a powerful post-processing or enhancement component in synthetic or manipulated image pipelines, and its outputs exhibit characteristic artifacts that are relevant for forensic detection.

**Diffusion and latent diffusion models.** **ADM** (Dhariwal & Nichol, 2021) refines DDPMs by learning variances and using improved sampling, achieving competitive likelihoods and sample quality while reducing the number of denoising steps needed for high-fidelity images. **Stable Diffusion** (Rombach et al., 2022) is a latent diffusion model (LDM) that performs the diffusion process in a compressed latent space rather than pixel space, enabling efficient high-resolution text-to-image generation on consumer GPUs. It has become a widely used open-source backbone for downstream fine-tuning and domain adaptation. **GLIDE** (Nichol et al., 2022) is an early text-conditional diffusion model that explores classifier-based and classifier-free guidance, demonstrating strong photorealism and text alignment along with image editing capabilities.

**Industrial and specialized text-to-image systems.** **Midjourney** (Midjourney, 2022) is a proprietary large-scale text-to-image model served via a cloud API and Discord interface. It is trained on web-scale image–text data and optimized for artistic, stylized, and photorealistic generations; although its architecture and training data are not publicly documented, it constitutes an impor-

Table 10: Detection accuracy (ACC, %) of baseline diffusion detectors, DRCT, and our HSIC-based CLIP detectors on the DRCT-2M benchmark proposed in DRCT (Chen et al., 2024). *Ours* denotes our HSIC-bottleneck model on final-layer CLIP features, while *Ours w/ intermediate* further incorporates intermediate CLIP features, yielding consistently higher and near-saturated accuracy across all diffusion variants.

| Dataset | CNNSpot | F3Net | CLIP/RN50 | Conv-B | UnivFD | DIRE | DRCT | Ours | Ours w/ intermediate |
|---|---|---|---|---|---|---|---|---|---|
| LDM | 99.87 | 99.85 | 99.00 | 99.97 | 98.30 | 54.62 | 99.66 | 99.07 | 99.96 |
| SDv1.4 | 99.91 | 99.78 | 99.99 | 100.00 | 96.22 | 75.89 | 98.56 | 99.07 | 99.70 |
| SDv1.5 | 99.90 | 99.79 | 99.96 | 99.97 | 96.33 | 76.04 | 98.48 | 99.07 | 99.67 |
| SDv2 | 97.55 | 88.66 | 94.61 | 95.84 | 93.83 | 99.87 | 99.85 | 99.06 | 99.97 |
| SDXL | 66.25 | 55.85 | 62.08 | 64.44 | 91.01 | 59.90 | 96.10 | 99.06 | 99.97 |
| SDXL-Refiner | 86.55 | 87.37 | 91.43 | 82.00 | 93.91 | 93.08 | 98.68 | 99.06 | 99.91 |
| SD-Turbo | 86.15 | 68.29 | 83.57 | 80.82 | 86.38 | 99.77 | 99.59 | 99.07 | 99.97 |
| SDXL-Turbo | 72.42 | 63.66 | 64.40 | 60.75 | 85.92 | 57.55 | 83.30 | 99.07 | 99.97 |
| LCM-SDv1.5 | 98.26 | 97.39 | 98.97 | 99.27 | 90.44 | 87.29 | 98.45 | 99.06 | 99.97 |
| LCM-SDXL | 61.72 | 54.98 | 57.43 | 62.33 | 88.99 | 72.53 | 93.78 | 99.06 | 99.97 |
| SDv1-Ctrl | 97.96 | 97.98 | 99.74 | 99.80 | 90.41 | 67.85 | 96.68 | 99.07 | 99.96 |
| SDv2-Ctrl | 85.89 | 72.39 | 80.69 | 83.40 | 81.06 | 99.69 | 99.85 | 99.06 | 99.97 |
| SDXL-Ctrl | 82.84 | 81.99 | 82.03 | 73.28 | 89.06 | 64.40 | 97.66 | 99.06 | 99.97 |
| Avg. | 87.33 | 82.15 | 85.68 | 84.76 | 90.92 | 77.57 | 96.97 | 99.06 | 99.92 |

tant real-world generator in current synthetic media ecosystems. **VQDM** (Gu et al., 2022) targets efficiency rather than raw generation quality: it compresses large text-to-image diffusion models such as SDXL and SDXL-Turbo using vector quantization while preserving image quality and text alignment, making large generators more deployable in resource-constrained settings. **Wukong** (Wukong, 2022) in our context refers to the Chinese text-to-image diffusion line built on the Wukong-Huahua foundation model. Wukong-Huahua is a diffusion-based generator trained on the large-scale Chinese multimodal Wukong dataset (100M image–text pairs), targeting high-quality Chinese-language text-to-image synthesis and deployed on MindSpore / Ascend hardware.

Together, these models span a wide range of architectures (GANs vs. diffusion vs. latent diffusion), training paradigms (unconditional, conditional, and translation-based), and deployment regimes (open-source vs. proprietary, full-precision vs. compressed). Evaluating synthetic image detectors across this diverse set is crucial for assessing robustness and cross-generator generalization.

### A.16 COMPARISON ON THE DRCT-2M BENCHMARK

To assess whether our method remains effective on large-scale diffusion-generated image detection, we evaluate on the DRCT-2M benchmark introduced by DRCT (Chen et al., 2024). DRCT-2M contains a diverse collection of synthetic images generated from multiple Stable Diffusion families and variants, including LDM, SDv1.x/SDv2, SDXL, refiner and turbo variants, LCM-based accelerations, and ControlNet-based pipelines. We follow the same training protocol to train on the real images of MSCOCO (Lin et al., 2015) and the synthetic images generated by SDV1.4, and then evaluate on different testing subsets on DRCT-2M. Table 10 reports the detection accuracy of classical CNN-based detectors (CNNSpot (Wang et al., 2020), F3Net (Qian et al., 2020), Conv-B (Liu et al., 2022)), CLIP-based detectors (CLIP/RN50 (Radford et al., 2021), UnivFD (Ojha et al., 2023)), the diffusion-specific DIRE (Wang et al., 2023a) and DRCT (Chen et al., 2024) baselines, and our HSIC-based CLIP detectors. While DRCT already outperforms prior baselines in average accuracy, our method (*Ours*) further improves performance across almost all diffusion variants, and the variant that additionally exploits intermediate CLIP features (*Ours w/ intermediate*) achieves near-saturated performance ($\approx 99.9\%$) on all generators. This demonstrates that our approach generalizes strongly to diverse diffusion architectures and settings, beyond the GAN-based benchmarks considered in our main experiments.

