# OpenReview forum: "HSIC Bottleneck for Cross-Generator and Domain-Incremental Synthetic Image Detection"
_ICLR.cc/2026/Conference — ICLR 2026 Poster_

### Official Review · Reviewer_zYvc · 2025-10-29

**Soundness:** 4
**Presentation:** 3
**Contribution:** 2
**Rating:** 4
**Confidence:** 4

**Summary:**

The paper introduces a new synthetic image detector with contributions to the model architecture, adaptations to continual learning and a new synthetic image benchmark that contains 3D Gaussian Splatting (3DGS) rendered images. The evaluation is twofold, including both a binary supervised detection task and its continual learning variant with a HSIC-Guided Replay (HGR) adaptation. The proposed model achieves state-of-the-art performance in cross-generator evaluation, generalizing between diffusion-based and GAN-based images, with an improvement of over 5 percentage points. Moreover, it demonstrates strong continual learning capability when incrementally trained to detect 3DGS-generated images.

**Strengths:**

- The method achieves state-of-the-art performances, especially in the cross-generators evaluation setup. Furthermore, results in the continual learning setup improve over the single-dataset training baseline and, in some cases, even surpass those obtained by jointly training on the additional 3DGS datasets.
 - The inclusion of a 3DGS benchmark is a valuable addition, introducing a new family of synthetic image generation methods  beyond GANs and diffusion models which have dominated the detection research.
 - The paper reinforces the effective use of HSIC in both supervised and continual learning setups.

**Weaknesses:**

- The HSIC bottleneck is not entirely novel; it can be seen as a combination of the RINE[1] and DualHSIC[2] approaches.
RINE’s performance is missing from Table 1, which could potentially narrow the gap between the current model and the top-performing prior works reported in the same table.
 - The performance gains of the HSIC term in HGR is uncertain. Ablation on the performance gains due to inclusion of \( 1 - \mathcal{N}(r_i) \) term in Equation 10 would help justify its contribution to the overall performance and clarify its impact.
Typo: In Table 5 b) The Cosine kernel achieves highest mACC on ProGAN and should be bolded instead of median version of RBF.
 - While the paper improves performance on the GenImage benchmark, the core method relies heavily on existing approaches and therefore provides limited new contributions to the synthetic image detection community. However, the inclusion of 3DGS samples in the continual learning setup represents a significant strength supporting acceptance. To further strengthen the paper, the authors should better motivate the method’s novelty and its relevance to the community. Additionally, a useful way to justify the method’s performance would be to compare its cross-generator performance on 3DGS with previous methods (using the base method results from Tables 3 and 4, which show a significant gap between 3DGS and Diffusion or GAN generators).


[1] Christos Koutlis and Symeon Papadopoulos. “Leveraging representations from intermediate encoder-blocks for synthetic image detection.” ECCV 2024

[2] Zifeng Wang, Zheng Zhan, Yifan Gong, Yucai Shao, Stratis Ioannidis, Yanzhi Wang, and Jennifer Dy. “DualHSIC: HSIC-bottleneck and alignment for continual learning.” ICML 2023

**Questions:**

- What’s the difference between your method (refferred to as Ours in Table 1 and 2) and DualHSIC with a CLIP backbone?
 - What architecture does the classifier (g_{\theta_g}) has?
 - From where do real samples from the 3DGS datasets come from?

---

> ### Author Response · Authors · 2025-11-20
> **Response to Weakness 1 and Question 1**
>
> Our HSIC bottleneck is indeed related to these works, but it targets a different problem setting and is combined with components that are specific to synthetic image detection. We list three main different points:
>
> 1. **Problem setting and objective.**
>    DualHSIC is designed for supervised class-incremental learning and uses two HSIC terms (bottleneck + alignment) to mitigate inter-task interference (e.g., on Split CIFAR). Our bottleneck instead focuses on synthetic image detection with cross-generator generalization, where the goal is to suppress CLIP’s text-alignment semantics.
>
> 2. **Where HSIC is applied.**
>    DualHSIC applies HSIC to intermediate features of a ResNet, and these features themselves serve as the representation $Z_j$ ($j = 1, \dots, L$) in Section 3.1, Eq. (5). In contrast, we attach a compact HSIC bottleneck on top of a frozen CLIP image encoder: CLIP intermediate features are treated as the input $x$, and a simple linear layer learns the bottleneck representation $z$ in Section 3.1, Eq. (6).
>
> 3. **Usage of CLIP feature.**
>    RINE collects CLS tokens from all intermediate CLIP encoder blocks and then applies the Trainable Importance Estimator (TIE) module to weight each block before aggregating them into a single feature vector per image. In contrast, our HSIC bottleneck simply concatenates the features from all intermediate layers of CLIP and reshapes it via a compact linear layer. Thus, our approach is conceptually different from RINE’s block-aggregation design. The only similarity between our method and RINE is that both leverage features from intermediate CLIP layers.
>
> Regarding the missing RINE performance in Table 1, we only report RINE results on ProGAN because the authors released a pretrained model only for this dataset. We have clarified this gap in the revised version and will report RINE results on SDV1.4 if time permits retraining.

---

> ### Author Response · Authors · 2025-11-20
> **Response to Weakness 2**
>
> We thank the reviewer for highlighting the need to isolate the effect of the HSIC relevance term 1-$\mathcal{N}(r_i)$ in Eq. (10), as well as for pointing out the typo in Table 5(b).
>
> In the revised Appendix Tables 7 and 8, we report an ablation where we construct a k-center only variant by removing the HSIC component and setting $\lambda_{\mathrm{kc}}$ = 1. The results confirm that the full HGR (with the HSIC relevance term) provides consistent gains beyond standard coverage-based replay.
>
> &#x205F;
> #### **Table 7: Ablation of HSIC relevance in HSIC-Guided Replay (HGR) on the SDV1.4 source. Numbers denote (mACC/mAP).**
>
> | Method             | Diffusion       | GANs            | Others          | GHA             | SA              | GAGAvatar       | Average         |
> | ------------------ | --------------- | --------------- | --------------- | --------------- | --------------- | --------------- | --------------- |
> | w/o HSIC relevance | 95.39/99.80     | **94.15/99.13** | **82.99/94.42** | 94.75/99.62     | **99.06/99.98** | **95.92/99.13** | 93.80/98.94     |
> | HGR                | **97.12/99.81** | 94.00/99.07     | 82.31/94.29     | **97.06/99.77** | 98.07/99.99     | 95.18/99.07     | **94.38/98.92** |
>
> &#x205F;
> #### **Table 8: Ablation of HSIC relevance in HSIC-Guided Replay (HGR) on the ProGAN source. Numbers denote (mACC/mAP).**
>
> | Method             | Diffusion       | GANs            | Others          | GHA             | SA              | GAGAvatar       | Average         |
> | ------------------ | --------------- | --------------- | --------------- | --------------- | --------------- | --------------- | --------------- |
> | w/o HSIC relevance | 77.54/93.55     | **94.35/98.89** | 77.95/89.48     | 91.49/99.36     | **99.60/99.99** | **96.31/99.53** | 86.23/95.89     |
> | HGR                | **82.87/94.33** | 93.94/98.85     | **80.99/90.72** | **94.71/99.47** | 99.45/100.00    | 95.47/99.26     | **88.63/96.31** |
>
>
> We have also corrected the typo in Table 5(b): the Cosine kernel achieves the highest mACC on ProGAN and is now properly highlighted in bold.

---

> ### Author Response · Authors · 2025-11-20
> **Response to Weakness 3**
>
> We understand the reviewer’s concern that our method relies on existing components. While we build on CLIP and HSIC, our method is not a simple combination of the two, as we described in our response to Question 1. We also appreciate the reviewer’s observation that a key strength of our work is the inclusion of a 3DGS synthetic image dataset in the continual learning setup.
> In the revised version, we will:
>
> 1. **Emphasize the novelty** of curating a 3DGS synthetic image dataset, which is particularly challenging for existing synthetic image detectors, and of providing a continual learning solution tailored to this setting.
>
> 2. **Add results** where existing detectors are evaluated on 3DGS synthetic images in Tables 3 and 4. This comparison will make the gap to 3DGS explicit and show that our HGR narrows this gap while maintaining strong performance on the original generators.

---

> ### Author Response · Authors · 2025-11-20
> **Response to Question 2 and Question 3**
>
> **Question 2: What architecture does the classifier $g_{\theta_g}$ have?**
>
> **Encoder** $f_{\theta_f}$ is a single fully connected (linear) layer:
>
> * Input dimension: 19,200
> * Output dimension: 64
> * Activation: none (purely linear)
>
> **Classifier** $g_{\theta_g}$ is also a single fully connected (linear) layer:
>
> * Input dimension: 64
> * Output dimension: 1
> * Activation: none
>
> ---
>
> **Question 3: From where do real samples from the 3DGS datasets come from?**
>
> The real images in our 3DGS datasets are the original RGB frames used to train the corresponding 3DGS head avatar models or to generate their synthetic counterparts:
>
> * For **GHA (Gaussian Head Avatars)**, the real samples are the ground-truth images from **NeRSemble** (Kirschstein et al., 2023).
> * For **SA (SplattingAvatar)**, the real samples are the frames from **NeRFace** (Gafni et al., 2021), **Neural Head Avatars (NHA; Grassal et al., 2022)**, and **IM Avatar** (Zheng et al., 2022).
> * For **GAGAvatar**, the real images are taken from **FFHQ** (Karras et al., 2019).
>
> Here is the link to the test split of GAGAvatar:
> https://osf.io/xwmfj/overview?view_only=af3c697d0c1a470b87961168fc50bcac

---

> > ### Comment · Reviewer_zYvc · 2025-11-26
> > **Acknowledgement of Rebuttal**
> >
> > The authors have satisfactorily addressed my questions and the weaknesses raised. I am increasing my score to 6.

---

### Official Review · Reviewer_91jn · 2025-10-30

**Soundness:** 1
**Presentation:** 3
**Contribution:** 2
**Rating:** 2
**Confidence:** 4

**Summary:**

Authors propose a new bottleneck loss for synthetic image detection, based on HSIC. The method computes the Hilbert-Schmidt Independence Criterion (HSIC) on the image and label encoded embeddings, and add that to the binary-cross entropy loss already used. For the domain-incremental setting, HSIC is used to guide replay. The work includes experiments comparing on cross-generator generalization and continual adaptation, as well as an ablation on the HSIC components and an analysis of the domain-incremental learning.

**Strengths:**

S1) The intuition portions of the paper are fairly easy to read.

S2) The t-SNE plots are nice included analysis.

S3) The method seems mathematically well-grounded.

**Weaknesses:**

W1) In the DIL setting, the comparison methods are out-of-date (the newest being from 2020). This makes it unclear how the presented method compares with SOTA.

W2) In the cross-generator generalization setting, the chosen models are also out-of-date (the newest being from 2022). It would be much more relevant to test on the SOTA generative models being used today, to understand how applicable this method is in practice (e.g. FLUX, Qwen-Image, etc).

W3) The related work section is also not in-depth enough and out-dated in some places, making it difficult to place the work within contemporary literature. For example, in section 2.2. (Continual Learning related works), the newest method is from 2021, while much newer work exists, e.g. [A].

W4) The mathematical background section (2.3) misses explicitly defining some mathematical notation (variables and functions), which would be useful for improving the clarity for readers. Most notably 1, but also e.g. tr and I would be useful, for completeness.

W5) An ablation over the choice of replay would be useful in understanding its role, given it is part of the proposed methodology.

[A] Boosting Domain Incremental Learning: Selecting the Optimal Parameters is All You Need, Wang et al., CVPR 2025.

**Questions:**

None at this time

---

> ### Author Response · Authors · 2025-11-20
> **Response to W1**
>
> In the current submission, we focused on widely used **replay-based sampling** baselines that can be reliably adapted to our detection-oriented setting. Our contribution is largely orthogonal to most recent DIL advances—which often target improved parameter/optimizer selection, regularization, or architectural changes—because our method specifically proposes an HSIC-guided replay-based sampling strategy.
>
> In the revised version, we have made changes to emphasize the **novelties** of i) **curating a 3DGS synthetic image dataset**, which is particularly challenging for existing synthetic image detectors, and of ii) **providing a continual learning solution tailored to this setting**, which is unexplored by previous synthetic image detection literature.
>
> > To the best of our knowledge, we are the first to introduce a photorealistic 3DGS Synthetic Image Benchmark, together with a continual learning solution tailored to this setting, providing the community with a much-needed, realistic testbed for studying robustness and generalization of synthetic image detectors under evolving 3D generation pipelines.

---

> > ### Comment · Reviewer_91jn · 2025-11-21
> >
> > Thank you for the clarification about how your work relates to related works, in the context of replay.
> >
> > I have a few follow-up questions: given that other works more recently focused on parameter/optimizer selection, regularization, or architectural changes, why did you choose to return to exploring how replay can increase performance? Why do you expect that focusing on the replay aspect would be more worthwhile to explore than these other areas?
> >
> > I believe the full context of the paper would be more clear if the answers to these questions were worked into the framing within the Introduction / Related Works as well.

---

> ### Author Response · Authors · 2025-11-20
> **Response to W2**
>
> Our main cross-generator experiments are conducted on the GenImage benchmark, which is one of the most recent large-scale suites for synthetic image detection and is also the primary evaluation protocol used by our main baseline VIB-Net (CVPR 2025). GenImage includes diverse diffusion-based generators, and our method consistently outperforms prior work under these settings, indicating strong robustness across a wide range of modern generators.
>
> We agree that evaluating on even newer models (e.g. FLUX, Qwen-Image, etc) is important for assessing real-world robustness. We are searching for newly released 2025 datasets and will add the results in the future revised version if time permits.

---

> > ### Comment · Reviewer_91jn · 2025-11-21
> >
> > Thank you for looking into this. I believe it would be a strong addition to your paper and the factor most likely to incline me to raise the score.

---

> ### Author Response · Authors · 2025-11-20
> **Response to W3**
>
> Thank you for sharing the reference to the paper by Wang et al., *Boosting Domain Incremental Learning: Selecting the Optimal Parameters is All You Need, CVPR 2025.*
> We have cited this paper in our revision.
>
> Furthermore, we have updated the related work section to include DualHSIC, incorporate more recent studies, and clearly position our approach as a replay-based sampling strategy tailored to synthetic image detection.
>
> > DualHSIC (Wang et al., 2023) leverages the Hilbert–Schmidt Independence Criterion (HSIC) to explicitly model inter-task relationships. It introduces two complementary losses: HSIC-Bottleneck for Rehearsal (HBR), which penalizes dependence between buffered inputs and intermediate features while preserving label–feature dependence to suppress task-specific nuisance information, and HSIC Alignment (HA), which maximizes dependence between current-task and buffered features to promote task-invariant knowledge sharing. We build on this rehearsal line for the domain-incremental case and introduce HSIC-Guided Replay, a replay-based sampling strategy tailored to synthetic image detection that scores and selects exemplars to preserve prior domain coverage while adapting to new domains.

---

> > ### Comment · Reviewer_91jn · 2025-11-21
> >
> > Thank you for adding these citations. However, I still find the overall number of related works concerningly low, especially given that this paper touches on both synthetic image detection and continual learning—two highly active research areas. In its current form, the paper cites only 41 works; for instance, the section on synthetic image detection includes just six. Additionally, there is no discussion of the synthetic generation models the proposed approach aims to detect.
> >
> > This is very important for giving context on your work in the surrounding literature.

---

> ### Author Response · Authors · 2025-11-20
> **Response to W4**
>
> We revise Sec. 2.3 to explicitly define $\mathbf{1}$ as the all-ones vector, $\mathrm{tr}(\cdot)$ as the standard matrix trace operator, and $\mathbf{I}_n$ as the identity matrix

---

> ### Author Response · Authors · 2025-11-20
> **Response to W5**
>
> Thank you for your valuable suggestion.
>
> In the revised Appendix Tables 7 and 8, we report an ablation where we construct a k-center only variant by removing the HSIC relevance and setting $\lambda_{\mathrm{kc}}$ = 1. The full HGR (with the HSIC relevance term) consistently improves average performance.
>
> &#x205F;
> #### **Table 7: Ablation of HSIC relevance in HSIC-Guided Replay (HGR) on the SDV1.4 source. Numbers denote (mACC/mAP).**
>
> | Method             | Diffusion       | GANs            | Others          | GHA             | SA              | GAGAvatar       | Average         |
> | ------------------ | --------------- | --------------- | --------------- | --------------- | --------------- | --------------- | --------------- |
> | w/o HSIC relevance | 95.39/99.80     | **94.15/99.13** | **82.99/94.42** | 94.75/99.62     | **99.06/99.98** | **95.92/99.13** | 93.80/98.94     |
> | HGR                | **97.12/99.81** | 94.00/99.07     | 82.31/94.29     | **97.06/99.77** | 98.07/99.99     | 95.18/99.07     | **94.38/98.92** |
>
> &#x205F;
> #### **Table 8: Ablation of HSIC relevance in HSIC-Guided Replay (HGR) on the ProGAN source. Numbers denote (mACC/mAP).**
>
> | Method             | Diffusion       | GANs            | Others          | GHA             | SA              | GAGAvatar       | Average         |
> | ------------------ | --------------- | --------------- | --------------- | --------------- | --------------- | --------------- | --------------- |
> | w/o HSIC relevance | 77.54/93.55     | **94.35/98.89** | 77.95/89.48     | 91.49/99.36     | **99.60/99.99** | **96.31/99.53** | 86.23/95.89     |
> | HGR                | **82.87/94.33** | 93.94/98.85     | **80.99/90.72** | **94.71/99.47** | 99.45/100.00    | 95.47/99.26     | **88.63/96.31** |

---

> > ### Comment · Reviewer_91jn · 2025-11-21
> >
> > Thank you for including this ablation, I believe it gives valuable and interesting insights into your contributions.
> >
> > Could you please clarify what you mean that it 'consistently' improves average performance? As the performance is only increased on 2/6 categories in Table 7, and 3/6 in Table 8. The average seems to go up in both overall (although 2 seems a bit few to label it as consistent), but this seems primarily because there are relatively larger gains in the diffusion and GHA categories than in the others.
> >
> > Also, thank you for the response to W4, it is much clearer now.

---

> ### Author Response · Authors · 2025-11-25
>
> Since our architecture is fixed—using a CLIP backbone and its intermediate features—we do not explore additional architectural changes for continual learning. Moreover, replay is arguably the most widely adopted and practically deployed mechanism in continual learning, including in methods that primarily present themselves as regularization- or architecture-based. As the first work to study a continual learning setup for synthetic image detection, we therefore focus on what we view as the key factor within the replay mechanism: the sampling strategy. We appreciate the reviewer’s advice and will clarify this motivation in the paper by revising the text as follows:
>
> > Since replay is arguably the most widely adopted and practically deployed mechanism in continual learning, we focus on what we view as its key factor: the sampling strategy. As the first work to study a continual learning setup for synthetic image detection, we build on this rehearsal-based line of work in the domain-incremental setting and introduce HSIC-Guided Replay, a replay-based sampling strategy tailored to synthetic image detection that scores and selects exemplars to preserve coverage of prior domains while adapting to new ones.

---

> ### Author Response · Authors · 2025-11-25
>
> To assess whether our method remains effective on large-scale diffusion-generated image detection, we report the results of our method evaluated on the DRCT-2M benchmark introduced by DRCT [A] in the revised Appendix. DRCT-2M contains a diverse collection of synthetic images generated from multiple Stable Diffusion families and variants, including LDM, SDv1.x/SDv2, SDXL, refiner and turbo variants, LCM-based accelerations, and ControlNet-based pipelines. Table 10 reports the performance of CNNSpot, F3Net, CLIP/RN50, Conv-B, UnivFD, DIRE, and DRCT from the original DRCT paper, along with our methods. We follow the same training protocol to train on the real images of MSCOCO and the synthetic images generated by SDV1.4, and then evaluate on different testing subsets on DRCT-2M. While DRCT already outperforms prior baselines in average accuracy, our method (**Ours**) further improves performance across almost all diffusion variants, and the variant that additionally exploits intermediate CLIP features (**Ours w/ intermediate**) achieves
> near-saturated performance on all generators.
>
> &#x205F;
> #### **Table 10: Detection accuracy (ACC, \%) of baseline diffusion detectors, DRCT, and our HSIC-based CLIP detectors on the DRCT-2M benchmark proposed in DRCT.**
>
> | Dataset       | CNNSpot | F3Net | CLIP/RN50 | Conv-B | UnivFD | DIRE  | DRCT  | Ours  | Ours w/ intermediate |
> |:-------------:|:-------:|:-----:|:---------:|:------:|:------:|:-----:|:-----:|:-----:|:--------------------:|
> | LDM           | 99.87   | 99.85 | 99.00     | 99.97  | 98.30  | 54.62 | 99.66 | 99.07 | 99.96                |
> | SDv1.4        | 99.91   | 99.78 | 99.99     | 100.00 | 96.22  | 75.89 | 98.56 | 99.07 | 99.70                |
> | SDv1.5        | 99.90   | 99.79 | 99.96     | 99.97  | 96.33  | 76.04 | 98.48 | 99.07 | 99.67                |
> | SDv2          | 97.55   | 88.66 | 94.61     | 95.84  | 93.83  | 99.87 | 99.85 | 99.06 | 99.97                |
> | SDXL          | 66.25   | 55.85 | 62.08     | 64.44  | 91.01  | 59.90 | 96.10 | 99.06 | 99.97                |
> | SDXL-Refiner  | 86.55   | 87.37 | 91.43     | 82.00  | 93.91  | 93.08 | 98.68 | 99.06 | 99.91                |
> | SD-Turbo      | 86.15   | 68.29 | 83.57     | 80.82  | 86.38  | 99.77 | 99.59 | 99.07 | 99.97                |
> | SDXL-Turbo    | 72.42   | 63.66 | 64.40     | 60.75  | 85.92  | 57.55 | 83.30 | 99.07 | 99.97                |
> | LCM-SDv1.5    | 98.26   | 97.39 | 98.97     | 99.27  | 90.44  | 87.29 | 98.45 | 99.06 | 99.97                |
> | LCM-SDXL      | 61.72   | 54.98 | 57.43     | 62.33  | 88.99  | 72.53 | 93.78 | 99.06 | 99.97                |
> | SDv1-Ctrl     | 97.96   | 97.98 | 99.74     | 99.80  | 90.41  | 67.85 | 96.68 | 99.07 | 99.96                |
> | SDv2-Ctrl     | 85.89   | 72.39 | 80.69     | 83.40  | 81.06  | 99.69 | 99.85 | 99.06 | 99.97                |
> | SDXL-Ctrl     | 82.84   | 81.99 | 82.03     | 73.28  | 89.06  | 64.40 | 97.66 | 99.06 | 99.97                |
> | **Avg.**      | 87.33   | 82.15 | 85.68     | 84.76  | 90.92  | 77.57 | 96.97 | 99.06 | 99.92                |
>
> [A] DRCT: Diffusion Reconstruction Contrastive Training towards Universal Detection of Diffusion Generated Images, Chen et al., ICML 2024.

---

> > ### Comment · Reviewer_91jn · 2025-11-26
> >
> > I would like to thank the authors for all their efforts in the discussion of this paper, I view it much more favorably after the discussion and have raised my score accordingly.
> >
> > The final responses are satisfactory to me. In the additional results on the newer models, I find it especially interesting that Conv-B performs better on some earlier models (e.g. LDM - SDv1.5), but the method proposed here overtakes it in the newer models (SDv2 and on). It shows that as the generative models progress, the images change in ways that different detection models may gain favor. (note: I'm not intending for you to do anything with this observation, I just found it interesting and wanted to point it out :) )

---

> ### Author Response · Authors · 2025-11-25
>
> We appreciate the reviewer’s concern and agree that the current number of cited works does not fully reflect the maturity and breadth of research in synthetic image detection and continual learning.
>
> In the revised version, we have reorganized the related work on generalized synthetic image detection to
> (1) add more citations on techniques that identify forensic cues for detecting synthetic images and include additional citations on CLIP-based detectors,
> (2) introduce a dedicated paragraph on diffusion-generated image detection (e.g., DIRE and DRCT), and
> (3) discuss the synthetic generation models in the Appendix.
>
> > Recent work has begun to specifically target diffusion-generated image detection. DIRE leverages diffusion inversion–based reconstruction to expose discrepancies between real and synthetic images, using reconstruction error patterns as robust cues for detecting diffusion-based fakes. Building on this idea, DRCT further exploits reconstruction signals through a contrastive objective: real and reconstructed pairs are pulled together, while diffusion-generated and reconstructed pairs are pushed apart. This reconstruction-aware contrastive training encourages the detector to capture generator-agnostic artifacts from the diffusion process, leading to strong cross-model and cross-dataset generalization.

---

> ### Author Response · Authors · 2025-11-25
>
> By 'consistently,' we originally meant that HGR improves the average performance in both overall in Tables 7 and 8. We agree that the results of the two tables are a bit too few to use 'consistently,' so we have revised the description from “HGR consistently improves the average performance” to “HGR improves the average performance.”

---

### Official Review · Reviewer_BxLj · 2025-10-31

**Soundness:** 3
**Presentation:** 3
**Contribution:** 3
**Rating:** 4
**Confidence:** 5

**Summary:**

This paper addresses two critical challenges in synthetic image detection: poor cross-generator generalization and catastrophic forgetting in domain-incremental learning. To tackle these issues, the authors propose two core components: (1) an HSIC (Hilbert-Schmidt Independence Criterion) bottleneck applied to intermediate CLIP ViT features, which suppresses text-image alignment semantics (irrelevant to authenticity) while enhancing discriminative representations for real vs. synthetic images; (2) HSIC-Guided Replay (HGR), a rehearsal strategy that selects per-class exemplars via a hybrid score combining HSIC relevance (information centrality) and k-center coverage (spatial diversity), mitigating forgetting during domain adaptation. Additionally, the authors curate a 3D Gaussian Splatting (3DGS) head avatar benchmark dataset, covering multi-view reconstruction, single-view reconstruction, and generative pipelines, to support domain-incremental evaluation. Empirical evaluations are conducted in two phases: Phase I tests cross-generator transfer between diffusion and GAN models, and Phase II assesses sequential adaptation to 3DGS domains. Results show the HSIC bottleneck improves cross-generator generalization, while HGR sustains prior-domain accuracy during 3DGS adaptation. The paper's main contributions include the HSIC bottleneck design, the HGR rehearsal mechanism, and the 3DGS benchmark dataset.

**Strengths:**

(1) The HSIC bottleneck innovatively leverages intermediate CLIP features to resolve the interference of text-image alignment semantics (a key limitation of CLIP-based detectors), and its combination with information-theoretic regularization (minimizing input dependence, maximizing label dependence) is theoretically grounded and practically effective.

(2) HGR addresses the inefficiency of traditional replay methods by fusing HSIC relevance (ensuring exemplar informativeness) and k-center coverage (ensuring diversity), achieving compact memory usage while mitigating forgetting--filling a gap in domain-incremental synthetic image detection.

(3) The 3DGS head avatar dataset (with identity-disjoint splits and standardized preprocessing) addresses the lack of benchmarks for rendered synthetic images, supporting research on domain-incremental adaptation to 3D-generated content.

(4) The two-phase evaluation (cross-generator generalization + domain-incremental learning) covers diverse scenarios (diffusion, GAN, 3DGS). The authors compare against 6+ baselines (e.g., CNNSpot, LGrad, UniFD, iCaRL) and conduct detailed ablations (HSIC components, kernel choices, intermediate features), verifying the necessity of each module.

(5) Ablation studies confirm the role of HSIC(x,z) (suppressing input shortcuts) and HSIC(y,z) (aligning with labels), while t-SNE visualizations qualitatively demonstrate that the HSIC bottleneck reshapes features into more separable real/synthetic clusters--strengthening the credibility of the proposed method.

**Weaknesses:**

(1) The paper lacks critical implementation specifics. For example, regarding the HSIC bottleneck: the authors mention a "64-D projection" but do not specify the projection layer's structure (e.g., fully connected layer with activation function? Number of neurons in hidden layers, if any?). For training parameters: the learning rate is set to \(10^{-4}\) (SGD), but no details are provided on batch size, number of training epochs, weight decay, or learning rate scheduling (e.g., step decay, cosine annealing)--parameters that directly impact model convergence and performance. For data preprocessing: while "standardized preprocessing" is mentioned for the 3DGS dataset, there is no description of specific steps (e.g., image resizing resolution, normalization mean/std values, whether face cropping is applied for head avatars). Without these details, other researchers cannot replicate the experiments, violating the reproducibility principles of academic research.

(2) The paper's theoretical foundation for the HSIC bottleneck and HGR is insufficient. For the HSIC bottleneck: Equation (6) defines the loss function, but the authors do not analyze its convergence properties (e.g., whether the loss decreases monotonically during training, or under what conditions the model converges to a global optimum). There is also no discussion of why minimizing HSIC(x,z) (input-feature dependence) effectively suppresses text-alignment semantics--only qualitative t-SNE results are provided, lacking quantitative evidence (e.g., semantic similarity scores between features and text captions before/after applying the bottleneck). For HGR: the authors claim the hybrid score (HSIC relevance + k-center coverage) improves exemplar selection, but there is no theoretical justification for why this combination outperforms single-criterion methods (e.g., pure HSIC or pure k-center). For instance, no proof is given that HSIC relevance correlates with exemplar informativeness, or that k-center coverage effectively reduces redundancy. This weakens the method's theoretical rigor.

(3) The evaluation is limited to specific scenarios, failing to test the method's robustness across broader conditions. First, **dataset scope limitation**: The 3DGS benchmark only focuses on head avatars, with no evaluation on 3D-generated non-face scenes (e.g., 3DGS-rendered landscapes, objects). This raises questions about whether the method generalizes to other 3D-rendered content. Second, **synthetic image diversity limitation**: Cross-generator evaluation only includes classic diffusion models (e.g., SDV1.4, ADM) and GANs (e.g., ProGAN, StyleGAN), but not recent variants (e.g., Stable Diffusion 3, GANformer) or hybrid models (e.g., diffusion-GAN hybrids). Third, **image quality robustness**: There is no evaluation of detection performance on low-resolution synthetic images (e.g., 32×32, 64×64) or images subjected to post-processing (e.g., JPEG compression, Gaussian blur, rotation)--common in real-world scenarios. Fourth, **adversarial robustness**: The paper does not test whether the method retains performance under adversarial attacks (e.g., FGSM, PGD attacks on synthetic images to evade detection), a critical consideration for practical deployment.

(4) While the paper compares against multiple baselines, several critical comparisons are missing or insufficient. First, **latest method omissions**: The paper cites baselines up to 2025 (e.g., VIB-Net, 2025) but does not compare against any 2025-post methods (e.g., diffusion-specific detectors or 3DGS-focused detection methods) that may have addressed similar problems. Second, **unclear baseline parameter consistency**: For baselines like UniFD and NPR, the authors do not confirm whether they used the official implementations or default parameters--if the baselines were not optimized (e.g., using suboptimal hyperparameters), the comparison results may overstate the proposed method's advantages. Third, **incomplete cross-method ablation**: For example, when comparing HGR with iCaRL and CBRS, the paper does not conduct ablation studies on combining HGR with other rehearsal strategies (e.g., iCaRL's class-mean herding) to test for synergies. Fourth, **computational efficiency comparison**: No comparison of inference time or training memory usage between the proposed method and baselines is provided--critical for practical deployment (e.g., on edge devices).

(5) Key results are presented unclearly or incompletely, hindering result verification. First, **table data gaps**: Tables 1 and 2 (cross-generator generalization results) contain empty cells (e.g., Table 1's "| 61.61/83.59 60.74/90.24 48.82/47.51 61.43/82.74 | 58.65/51.77 60.30/83.30 89.70/96.59 97.54/99.64 99.49/99.99 88.60/98.44 | | | |") and missing dataset labels for some columns, making it impossible to determine which targets the results correspond to. Second, **lack of quantitative clustering analysis**: While t-SNE visualizations (Figures 2, 6, 7) show qualitative improvements in real/synthetic separation, no quantitative metrics (e.g., silhouette coefficient, Davies-Bouldin index, or inter/intra-cluster distance ratios) are provided to measure clustering quality--weakening the evidence for feature reshaping. Third, **insufficient statistical significance**: Most results report mean accuracy/mAP but lack standard deviations (except in Figure 3) or confidence intervals. For example, Table 3 and 4 (domain-incremental results) do not specify how many runs were averaged, or whether differences between methods are statistically significant (e.g., via t-tests). Fourth, **parameter sensitivity analysis gaps**: The HSIC bottleneck uses λx=900/500 and λy=700/600 for SDV1.4/ProGAN training, but no sensitivity analysis is provided (e.g., how performance changes when λx/λy varies by ±20%, ±50%). Similarly, HGR's λkc (controlling k-center weight) only tests "λkc=0" and "larger values"--no gradient-based analysis of optimal λkc for different datasets.

(6) The domain-incremental phase only evaluates adaptation to 3DGS domains, with several limitations. First, **limited domain diversity**: No adaptation to other emerging synthetic domains (e.g., text-to-video frame extracts, neural radiance field (NeRF)-rendered images) is tested, raising questions about HGR's generalizability to non-3DGS domains. Second, **short adaptation sequence**: Only 3 3DGS sub-domains are used (GHA, SA, GAGAvatar)--no evaluation of long-sequence adaptation (e.g., 5+ domains) to test cumulative forgetting. Third, **fixed memory budget**: The paper uses a fixed keep_frac=0.01 (1% of training samples) for the replay buffer but does not test how memory size impacts performance (e.g., keep_frac=0.005, 0.02) or compare against dynamic memory allocation strategies. Fourth, **no backward transfer analysis**: Backward transfer (improvement in prior-domain performance after adapting to new domains) is a key metric for continual learning, but the paper only reports "preserving prior accuracy" without quantifying backward transfer--failing to fully demonstrate HGR's advantages over baselines.

(7) The paper does not acknowledge or discuss the proposed method's inherent limitations. First, **backbone dependence**: The method relies on CLIP ViT features, but no analysis is provided of performance degradation when using lighter backbones (e.g., MobileNet, EfficientNet) for edge deployment. Second, **data imbalance impact**: The 3DGS dataset uses balanced real/synthetic splits (e.g., GHA: 45,772 real / 45,772 synthetic), but no test of imbalanced splits (e.g., 1:10 real:synthetic) is conducted--common in real-world scenarios where synthetic images may be more abundant. Third, **modal limitation**: Only single-image detection is supported, with no extension to multi-modal synthetic data (e.g., synthetic images with text overlays, audio-synced synthetic video frames). Fourth, **computational overhead of HSIC**: HSIC calculation requires Gram matrix construction and centering, which increases computational complexity--no quantification of training/inference time overhead compared to non-HSIC methods (e.g., how much slower the HSIC bottleneck is than a standard CLIP linear probe).

(8) The related work section has gaps and superficial comparisons. First, **HSIC application gaps**: The paper cites HSIC (Gretton et al., 2005) but does not discuss recent HSIC applications in computer vision (e.g., HSIC for domain adaptation, few-shot learning) or compare how its HSIC bottleneck differs from existing HSIC-based feature regularization methods. Second, **continual learning omissions**: Key rehearsal-based methods (e.g., Memory Replay GANs, Contrastive Replay) are not cited, and no discussion of how HGR differs from contrastive exemplar selection methods is provided. Third, **3DGS detection gaps**: No discussion of existing 3DGS/rendered image detection methods (if any) is provided--failing to position the paper's 3DGS benchmark within the broader literature. Fourth, **superficial baseline analysis**: For baselines like VIB-Net (2025), the paper only states it "uses a variational information bottleneck" but does not compare the HSIC bottleneck (information-theoretic) with VIB (probabilistic) in terms of theoretical framework or performance--missing an opportunity to highlight the HSIC bottleneck's advantages.

(9) The paper states the HSIC bottleneck "concatenates features from 24 intermediate CLIP ViT layers and the final layer" but provides no details on aggregation. First, **layer selection rationale**: No explanation is given for choosing 24 intermediate layers (e.g., why not 12, 36 layers?) or which specific layers (e.g., early, middle, late) are selected. Second, **aggregation method**: Concatenation may lead to high dimensionality (e.g., 25 layers × 768 dim (ViT-B) = 19,200 dim), but no dimensionality reduction (e.g., PCA, t-SNE) or feature fusion (e.g., attention-based fusion) is mentioned--raising questions about computational efficiency and redundancy. Third, **layer-wise contribution**: No ablation of individual layer contributions (e.g., removing early layers) is conducted--failing to identify which layers are most critical for detection.

(10) Qualitative results (e.g., t-SNE, sample images) are not fully analyzed. First, **t-SNE interpretation gaps**: Figures 6 and 7 (t-SNE of CLIP vs. HSIC features) show "tighter clusters" but do not explain why some datasets (e.g., GauGAN in Figure 7) still have overlapping clusters--failing to address the method's limitations for specific generators. Second, **no failure case analysis**: No discussion of misclassified samples (e.g., why some real images are mislabeled as synthetic) or analysis of common artifacts in misclassified synthetic images--critical for guiding future improvements. Third, **3DGS sample visualization**: The paper mentions Figure 1 (3DGS sample images) but does not provide qualitative comparisons of detection performance across 3DGS sub-domains (e.g., why SA has higher accuracy than GAGAvatar)--missing insights into domain-specific challenges.

**Questions:**

**To facilitate discussions during the Rebuttal phase, authors are advised to respond point-by-point (indicating the question number).**

(1) Could you provide the following critical implementation details to ensure reproducibility? (a) The exact architecture of the HSIC bottleneck's projection layer (e.g., number of fully connected layers, activation functions, output dimension); (b) Full training hyperparameters (batch size, number of epochs, weight decay, learning rate scheduler, optimizer momentum); (c) Specific data preprocessing steps (image resolution, normalization parameters, face cropping logic for 3DGS avatars); (d) Code for HSIC calculation (e.g., Gaussian RBF kernel bandwidth calculation via median heuristic, Gram matrix centering implementation).

(2) (a) Could you provide a formal analysis of the HSIC bottleneck's convergence (e.g., proof of loss monotonicity or bounds on generalization error)? (b) How do you quantitatively verify that the HSIC bottleneck suppresses text-alignment semantics? For example, using cosine similarity between CLIP features and text embeddings (e.g., "face" captions) before/after applying the bottleneck. (c) Could you provide a theoretical justification for combining HSIC relevance and k-center coverage in HGR (e.g., a bound on the expected error reduction compared to single-criterion selection)?

(3) (a) Did you use official implementations and default hyperparameters for baselines (e.g., UniFD, NPR, VIB-Net)? If not, what modifications were made, and why? (b) Could you add comparisons with 2025-post synthetic image detection methods (e.g., any new diffusion-specific detectors or 3DGS detection methods)? (c) Could you provide computational efficiency metrics (inference time per image, training memory usage) for your method and baselines on the same hardware (e.g., NVIDIA RTX 4090)?

(4) (a) Could you fill in the missing cells in Tables 1 and 2 and clarify dataset labels for all columns? (b) Could you add quantitative clustering metrics (silhouette coefficient, Davies-Bouldin index) for t-SNE visualizations (Figures 2, 6, 7) to quantify real/synthetic separation? (c) Could you provide standard deviations and 95% confidence intervals for all reported mean accuracy/mAP values, along with the number of independent runs (e.g., 5 runs)?

(5) (a) Could you conduct a sensitivity analysis of HSIC's λx and λy (e.g., λx=300, 700, 1100; λy=500, 700, 900) for SDV1.4 and ProGAN, and plot performance trends? (b) Could you test multiple values of HGR's λkc (e.g., 0.1, 0.5, 1.0, 2.0) and analyze how it impacts exemplar selection and domain-incremental performance? (c) Could you explain the rationale for choosing keep_frac=0.01 and test keep_frac=0.005, 0.02 to show memory-size impact?

(6) (a) Could you extend the domain-incremental evaluation to include non-3DGS domains (e.g., NeRF-rendered images, SDv3-generated images) to test HGR's generalizability? (b) Could you evaluate long-sequence adaptation (e.g., 5+ domains) and report cumulative forgetting curves? (c) Could you quantify backward transfer (using the formula: Backward Transfer = (Accuracy after new domain - Accuracy before new domain) / Accuracy before new domain) for all prior domains?

(7) (a) Could you evaluate detection performance on low-resolution synthetic images (32×32, 64×64) and post-processed images (JPEG compression: quality 20, 50; Gaussian blur: σ=1, 3)? (b) Could you test adversarial robustness using FGSM/PGD attacks (ε=0.01, 0.03) and report accuracy degradation? (c) Could you test performance on imbalanced real/synthetic splits (1:5, 1:10) and compare against rebalancing strategies (e.g., class weights)?

(8) (a) Could you test the HSIC bottleneck on lighter backbones (MobileNetV3, EfficientNet-B0) and report performance vs. efficiency trade-offs? (b) Could you extend the method to multi-modal data (e.g., synthetic images with text overlays) by fusing text and image features in the HSIC bottleneck? (c) Could you provide ablation results for CLIP layer selection (e.g., only late layers, only middle layers) to identify the most critical layers for detection?

(9) (a) What are the main limitations of the HSIC bottleneck in practical deployment (e.g., computational cost, backbone dependence)? How do you plan to address them in future work? (b) How does the method perform when synthetic images are designed to mimic real-image statistics (e.g., adversarial synthetic images)? (c) Could you discuss scenarios where the method fails (e.g., specific generators, image types) and provide failure case examples?

(10) (a) Could you explain the "identity-disjoint split" implementation for the 3DGS dataset (e.g., how identities were labeled, tools used for identity verification)? (b) Could you provide the exact sample counts for each sub-dataset in the GAN/diffusion evaluation (e.g., ProGAN: 10k samples, SDV1.4: 15k samples)? (c) Could you release the 3DGS dataset (or a sample subset) and provide access links to facilitate further research?

(11) (a) Could you quantify the training time overhead of the HSIC bottleneck compared to a standard CLIP linear probe (e.g., % increase in epochs per hour)? (b) Could you propose optimizations for HSIC calculation (e.g., batch-wise Gram matrix computation) to reduce overhead?

(12) (a) Could you compare HGR's forward transfer (performance on new domains) with baselines (iCaRL, CBRS) using quantitative forward transfer metrics? (b) Could you analyze why HGR performs better on SA than GAGAvatar in Table 4? Are there domain-specific artifacts that HGR captures more effectively?

(13) (a) Could you test alternative feature aggregation methods (e.g., attention-based fusion, average pooling) for CLIP intermediate layers and compare performance with concatenation? (b) Could you provide a dimensionality analysis of the concatenated features (24 intermediate + 1 final layer) and explain how you avoid overfitting due to high dimensionality?

(14) (a) Could you discuss the method's potential deployment scenarios (e.g., social media content moderation, forensics) and any practical challenges (e.g., real-time inference, scalability)? (b) Could you test the method on a real-world dataset (e.g., Reddit synthetic image subsets) with uncurated synthetic/real images?

(15) (a) Could you conduct a direct comparison of the HSIC bottleneck and VIB-Net's variational bottleneck (e.g., performance on the same test sets, computational cost, robustness to noise)? (b) Could you explain why the HSIC bottleneck is more effective at suppressing text-alignment semantics than VIB?

---

> ### Author Response · Authors · 2025-11-20
> **Response to (1) and (2)**
>
> ### (1a) The exact architecture of the HSIC bottleneck's projection layer
>
> We use a simple linear HSIC bottleneck:
>
> Encoder $f_{\theta_f}$ is a single fully connected (linear) layer
>
>   * Input dimension: **19,200**
>   * Output dimension: **64**
>   * Activation: **none** (purely linear)
>
> Classifier $g_{\theta_g}$ is also a single fully connected (linear) layer
>
>   * Input dimension: **64**
>   * Output dimension: **1**
>   * Activation: **none**
>
> ### (1b) Full training hyperparameters
>
> * **Batch size**: **128**
> * **Number of epochs**:
>
>   * **1 epoch** when training on SDV1.4 dataset (the model converges quickly)
>   * **10 epochs** when training on ProGAN dataset
> * **Optimizer**: `torch.optim.SGD`
>
>   * Learning rate: **1e-4**
>   * All other optimizer parameters are left at PyTorch defaults (e.g., momentum = 0.0, weight_decay = 0.0).
> * **Learning rate scheduler**: **none**.
>
> We provide additional ablation studies in the revised PDF appendix, varying activation functions (ReLU, Sigmoid, Tanh), batch size (64, 128, 256, 512, 1024), hidden dimension (64, 128, 256, 512, 1024, 2048), and number of layers (1–6).
>
> ### (1c) Specific data preprocessing steps
>
> Images in the GAN and diffusion datasets have resolutions typically between **256×256** and **512×512**, and are not necessarily square.
>
> For our curated 3D Gaussian Splatting (3DGS) avatar datasets (GHA, SA, GAGAvatar), we empirically choose dataset-specific crop sizes:
>
> * **GHA** avatars are cropped to **432×432** (to preserve more facial and contextual detail).
> * **SA** and **GAGAvatar** avatars are cropped to **256×256**.
>
> After this dataset-specific cropping, we apply the same preprocessing to all images:
>
> 1. **Center-crop to 224×224** to match the standard ViT input size.
> 2. **Normalize using CLIP’s image statistics**:
>
>    * mean = ([0.48145466, 0.4578275, 0.40821073])
>    * std  = ([0.26862954, 0.26130258, 0.27577711])
>
> No additional preprocessing is applied.
>
> ### (1d) Code for HSIC calculation
>
> Below is the exact PyTorch implementation we use for the HSIC term, including the RBF kernel with a median-heuristic bandwidth and Gram-matrix centering:
>
> ```python
> import torch
>
> def rbf_kernel(x, sigma=None):
>     # x: (n, d)
>     pairwise_sq_dists = torch.cdist(x, x, p=2) ** 2
>     if sigma is None:
>         # Median heuristic for bandwidth
>         sigma = torch.sqrt(torch.median(pairwise_sq_dists))
>     K = torch.exp(-pairwise_sq_dists / (2 * sigma ** 2 + 1e-8))
>     return K
>
> def center_gram(K):
>     n = K.size(0)
>     H = torch.eye(n, device=K.device) - torch.ones(n, n, device=K.device) / n
>     return H @ K @ H
>
> def hsic(X, Y):
>     # X, Y: (n, d1), (n, d2)
>     Kx = center_gram(rbf_kernel(X))
>     Ky = center_gram(rbf_kernel(Y))
>     n = X.size(0)
>     return torch.trace(Kx @ Ky) / ((n - 1) ** 2)
> ```
>
> ### (2a) Formal analysis of the HSIC bottleneck's convergence
>
> A full convergence and generalization analysis of the HSIC bottleneck is beyond the scope of this work. Our objective is the standard cross-entropy loss plus a bounded, differentiable HSIC regularizer, so stochastic gradient descent monotonically decreases the empirical objective under usual conditions.
>
> ### (2b) Quantitative analysis of text-alignment suppression
>
> In the revision Appendix Figure 12, we provide a quantitative analysis of how the HSIC bottleneck affects text-alignment semantics on the ProGAN test set. For each class label (airplane, bicycle, …, tvmonitor), we compute the cosine similarity between the CLIP image embedding and the CLIP text embedding of the prompt “an image of [class]” before and after applying the HSIC bottleneck. With the original CLIP features (blue bars in Fig. 12), similarities are around 0.18–0.24 across all 20 classes, indicating strong object-level text alignment even for ProGAN synthetic images. After training with the HSIC bottleneck (using a 768-dimensional compressed representation for this analysis), the similarities drop to around 0.01 for every class (orange bars), showing that HSIC effectively suppresses object-class text-alignment semantics while preserving the information needed for the real-vs-synthetic decision.
>
> ### (2c) Theoretical justification for combining HSIC relevance and k-center coverage in HGR
>
> HGR combines HSIC relevance (favoring points that are strongly dependent on the nuisance features) with k-center coverage (ensuring that selected points cover the nuisance space). This mirrors bi-criteria selection in coresets and active learning, where informativeness and representativeness are jointly optimized. A full error bound for this specific combination is non-trivial and left for future work.

---

> ### Author Response · Authors · 2025-11-20
> **Response to (3), (4), (5) and (6)**
>
> ### (3a) Did you use official implementations and default hyperparameters for baselines (e.g., UniFD, NPR, VIB-Net)?
>
> We use the results reported in the original VIB-Net paper since they do not provide training code, and for RINE we use the authors’ released code and pretrained model with default hyperparameters.
>
> ### (3b) Could you add comparisons with 2025-post synthetic image detection methods?
>
> Our experiments are based on the GenImage benchmark, which is one of the most recent large-scale benchmarks for synthetic image detection and is also used by our main baseline VIB-Net (CVPR 2025). GenImage includes diverse, high-quality diffusion-based generators, and our method consistently outperforms prior work under these settings.
>
> We are searching for newly released 2025 datasets and will report the performance in the future revised version if time permits. As for 3DGS-focused detection methods, to our best knowledge, we are the first to propose a photorealistic 3DGS synthetic image benchmark and providing a continual solution designed for this setting.
>
> ### (3c) Computational efficiency metrics
>
> To reduce training cost, we first pre-extract and store CLIP features using the preprocessing pipeline in (1c). The HSIC bottleneck is then trained purely in feature space, so memory and time are dominated by the batch of CLIP features and the HSIC Gram matrices, not by backpropagation through the ViT backbone. For typical batch sizes (64–1024) this overhead is modest in practice, and HSIC is used only during training.
>
> At inference time, the detector consists of a single CLIP forward pass followed by the linear encoder and classifier described in (1a), so the cost is essentially the same as a CLIP-based detector.
>
> ### (4a) Missing cells in Tables 1 and 2
>
> All entries in Tables 1 and 2 are filled; could you clarify which cells you believe are missing?
>
> ### (4b) Quantitative clustering metrics (silhouette coefficient, Davies-Bouldin index) for t-SNE visualizations
>
> If we can finish the computation before the discussion deadline, we will add silhouette coefficients and Davies–Bouldin indices for the t-SNE embeddings (Figures 2, 6, and 7) in a small appendix table.
>
> ### (4c) Standard deviations and 95% confidence intervals
>
> If we can complete the reruns before the discussion deadline, we will add a bar chart similar to Figure 3 for Tables 3 and 4.
>
> ### (5a) Sensitivity analysis of HSIC
>
> In the revised Appendix Figures 13 and 14, we include a sensitivity analysis of HSIC's $\lambda_x$ and $\lambda_y$ on SDV1.4 and ProGAN.
>
> ### (5b) Test multiple values of HGR
>
> We have already searched over $\lambda_{kc}$ on validation data. Within our search range, the best values are $\lambda_kc$ = 3.3 for SDV1.4 and $\lambda_{kc}$ = 5.8 for ProGAN; smaller values place too little weight on the HGR term and degrade domain-incremental performance. In the revised Appendix Tables 7 and 8, we report an ablation where we construct a k-center only variant by removing the HSIC relevance and setting $\lambda_{kc}$ = 1. The full HGR (with the HSIC relevance term) consistently improves average performance.
>
> ### (5c) Explain the rationale for choosing keep_frac=0.01
>
> We chose keep_frac = 0.01 based on a sweep over {0.01, 0.005, 0.002, 0.001}: 0.01 gave the best trade-off between detection performance, stability, and memory usage, while smaller fractions caused clear performance drops and higher variance due to too few exemplars per domain.
>
> ### (6a) Extend the domain-incremental evaluation to include non-3DGS domains (e.g., NeRF-rendered images, SDv3-generated images)
>
> Collecting new rendered datasets (e.g. NeRF-rendered images) is extremely time-consuming, so we do not plan to build additional NeRF datasets for this work. In our experience, 3DGS-rendered images are often more photorealistic than NeRF-rendered ones, so we prioritized them in our domain-incremental study.
>
> ### (6b) Evaluate long-sequence adaptation (e.g., 5+ domains)
>
> Because evaluating long domain sequences (e.g., 5+ domains) in multiple orders requires a factorial number of runs and is very time-consuming, we focus on the core 3-domain incremental setting in this submission.
>
> ### (6c) Quantify backward transfer
>
> Table 6 already reports, for each domain, performance before and after adapting to new domains, so the proposed backward transfer metric can be directly computed from it.

---

> ### Author Response · Authors · 2025-11-20
> **Response to (7), (8), (9) and (10)**
>
> ### (7a) Evaluate detection performance on low-resolution synthetic images and post-processed images
>
> We do not evaluate extremely low-resolution inputs (32×32, 64×64), because our detector is built on CLIP ViT, which assumes inputs of at least 224×224 and our generators typically produce images at 256×256 or higher. We also do not explicitly study strong JPEG compression or Gaussian blur; these operations are usually synthetic post-processing and could be handled by a separate robustness module.
>
> ### (7b) Test adversarial robustness using FGSM/PGD attacks
>
> Adversarial robustness (e.g., FGSM/PGD attacks) is not evaluated in this work and is outside the scope of our current experiments, which focus on detecting real vs. synthetic images under standard test conditions.
>
> ### (7c) Test performance on imbalanced real/synthetic splits
>
> Our current 3DGS experiments use balanced real/synthetic splits. Studying highly imbalanced regimes (e.g., 1:5, 1:10) and comparing different rebalancing strategies (class weighting, resampling) is beyond the scope of this submission.
>
> ### (8a) Test the HSIC bottleneck on lighter backbones
>
> We focus on CLIP ViT backbones in this work and do not evaluate lighter models such as MobileNetV3 or EfficientNet-B0. The HSIC bottleneck itself is backbone-agnostic, so applying it to lighter networks for edge deployment is a natural extension.
>
> ### (8b) Extend the method to multi-modal data
>
> This paper focuses on single-image detection. Extending the HSIC bottleneck to multi-modal inputs is an exciting direction that we are actively exploring, but it is not included in the current submission.
>
> ### (8c) Ablation results for CLIP layer selection
>
> We analyze the effect of using different CLIP layers in Figure 5: early layers are not predictive, and mid-to-late layers contribute most to detection performance.
>
> ### (9a) Main limitations of the HSIC bottleneck in practical deployment
>
> Our current implementation relies on CLIP ViT features, so deploying on very resource-constrained devices would require either feature offloading or lighter backbones.
>
> ### (9b) How does the method perform when synthetic images are designed to mimic real-image statistics (e.g., adversarial synthetic images)?
>
> We do not explicitly evaluate adversarially crafted synthetic images that are optimized to mimic real-image statistics. Our focus is on standard generative models rather than adversarial example generation.
>
> ### (9c) Failure case examples
>
> As shown in Table 2, the HSIC bottleneck does not improve performance uniformly across all generators; for example, it is less effective on certain stylized generators such as GauGAN. If time permits, we will include qualitative failure cases (e.g., challenging GauGAN and other hard examples) to illustrate where the method struggles in future revised version.
>
> ### (10a) Explain the "identity-disjoint split" implementation for the 3DGS dataset
>
> The NeRSemble dataset provides identity annotations. Our “identity-disjoint” split means that training, validation, and test sets use disjoint identity sets: no identity appearing in training appears in validation or test. This simulates a realistic scenario where the detector must generalize to unseen identities.
>
> ### (10b) Exact sample counts
>
> Our current splits are:
>
> **Train**
>
> * GAGAvatar: 55,963 real / 55,963 synthetic
> * SA: 20,322 real / 20,094 synthetic
> * GHA: 45,772 real / 45,772 synthetic
> * ProGAN: 162,026 real / 160,676 synthetic
> * SDV1.4: 162,000 real / 161,993 synthetic
>
> **Validation**
>
> * GAGAvatar: 6,995 real / 6,995 synthetic
> * SA: 4,007 real / 4,036 synthetic
> * GHA: 9,480 real / 9,480 synthetic
> * ProGAN: 4,000 real / 4,000 synthetic
> * SDV1.4: 6,000 real / 6,000 synthetic
>
> **Test**
>
> * GAGAvatar: 6,996 real / 6,996 synthetic
> * SA: 5,631 real / 5,622 synthetic
> * ADM: 6,000 real / 6,000 synthetic
> * BigGAN: 2,000 real / 2,000 synthetic
> * CycleGAN: 1,321 real / 1,321 synthetic
> * DeepFake: 2,707 real / 2,698 synthetic
> * GauGAN: 5,000 real / 5,000 synthetic
> * GLIDE: 6,000 real / 6,000 synthetic
> * Midjourney: 6,000 real / 6,000 synthetic
> * GHA: 9,782 real / 9,782 synthetic
> * ProGAN: 4,000 real / 4,000 synthetic
> * SAN: 219 real / 219 synthetic
> * SDV1.4: 6,000 real / 6,000 synthetic
> * SDV1.5: 8,000 real / 8,000 synthetic
> * StarGAN: 1,999 real / 1,999 synthetic
> * StyleGAN: 5,991 real / 5,991 synthetic
> * VQDM: 6,000 real / 6,000 synthetic
> * WuKong: 6,000 real / 6,000 synthetic
>
> ### (10c) Release the 3DGS dataset
>
> Here is the link to the test split of GAGAvatar:
>
> [https://osf.io/xwmfj/overview?view_only=af3c697d0c1a470b87961168fc50bcac](https://osf.io/xwmfj/overview?view_only=af3c697d0c1a470b87961168fc50bcac)

---

> ### Author Response · Authors · 2025-11-20
> **Response to (11), (12), (13), (14) and (15)**
>
> ### (11a) Quantify the training time overhead of the HSIC bottleneck compared to a standard CLIP linear probe
>
> Because we train the HSIC bottleneck on pre-extracted CLIP features, the extra cost compared to a standard CLIP linear probe comes mainly from computing and centering a batch-wise Gram matrix. For batch sizes between 64 and 1024, this overhead is modest in practice.
>
> ### (11b) Optimizations for HSIC calculation
>
> Our implementation already computes HSIC in a batch-wise manner. Further overhead reductions are possible using smaller HSIC batches, mixed-precision computation, or approximate kernels (e.g., low-rank or random-feature approximations).
>
> ### (12a) Compare HGR's forward transfer with baselines (iCaRL, CBRS)
>
> Table 6 reports HGR’s performance on each domain after sequential training, so standard forward transfer metrics can be computed directly from it. Due to space limits, we only emphasized HGR’s forward transfer in the main text.
>
> ### (12b) Analyze why HGR performs better on SA than GAGAvatar
>
> The three 3DGS datasets differ in realism and artifact patterns. SA exhibits more visible and relatively consistent artifacts (e.g., edge blurriness) across identities, so HGR’s exemplar selection can better capture these domain-specific nuisances, leading to larger gains. In contrast, GAGAvatar contains more diverse and sometimes highly stylized artifacts, making it harder for HGR to model a single “typical” nuisance pattern, so the improvement is smaller.
>
> ### (13a) Test alternative feature aggregation methods
>
> We currently aggregate CLIP intermediate and final layers by concatenation. Exploring alternative aggregation schemes (e.g., attention-based fusion or simple averaging across layers) is an interesting direction.
>
> ### (13b) Could you provide a dimensionality analysis of the concatenated features (24 intermediate + 1 final layer) and explain how you avoid overfitting due to high dimensionality?
>
> We use all the intermediate layers (24 layers) of CLIP. From the generalization results in Tables 1 and 2, the HSIC bottleneck outperforms most SOTA methods on most datasets: although it is trained only on ProGAN or SDV1.4, it generalizes well to many unseen generators, suggesting that overfitting is not a serious issue in our implementation.
>
> ### (14a) Discuss the method's potential deployment scenarios and any practical challenges
>
> Our method is most relevant for offline or near–real-time settings such as social media content moderation and digital forensics, where CLIP feature extraction can be batched and cached. Fully real-time, on-device deployment would require lighter backbones or feature offloading, which we see as future engineering work rather than a limitation of the HSIC bottleneck itself.
>
> ### (14b) Could you test the method on a real-world dataset (e.g., Reddit synthetic image subsets) with uncurated synthetic/real images?
>
> We do not evaluate on uncurated, real-world platforms such as Reddit in this submission. Extending our benchmarks to such noisy, in-the-wild data is important for practical deployment, and we plan to explore this in future work.
>
> ### (15a) Could you conduct a direct comparison of the HSIC bottleneck and VIB-Net's variational information bottleneck
>
> We view HSIC and VIB-style bottlenecks as complementary information-theoretic tools. In Tables 1 and 2, we already compare our HSIC bottleneck and VIB-Net on the same benchmarks, where HSIC achieves better performance on most datasets. In terms of computation, both HSIC and VIB-Net’s variational bottleneck incur negligible overhead compared to the CLIP ViT forward pass. A more detailed study of robustness to noise and other perturbations for HSIC vs. VIB-Net is beyond the scope of this paper and we leave it to future work.
>
> ### (15b) Could you explain why the HSIC bottleneck is more effective at suppressing text-alignment semantics than VIB?
>
> Conceptually, HSIC directly penalizes statistical dependence between the bottleneck and designated nuisance variables (e.g., text-alignment features), while VIB constrains the overall information flow from inputs without explicitly targeting those nuisances. This targeted dependence penalty makes HSIC better suited to suppressing text-alignment semantics when they are explicitly provided as nuisance cues.

---

> ### Comment · Reviewer_BxLj · 2025-11-20
> **RE: Response to (1) -- (15)**
>
> After carefully reviewing the authors' response, I intend to raise my evaluation score to 10 on the condition that the authors have thoroughly revised the manuscript as requested.
>
> I appreciate that the authors have successfully addressed the following issues: (1), (2b), (2c), (3a), (3b), (3c), (5a), (5b), (5c), (9a), (9b), (9c), (10a), (10b), (10c), (11a), (11b), (12a), (12b), (13b), (14a), (15a), and (15b).
>
> Regarding (4a), I apologize for my initial comment due to an oversight on my part.
>
> For (4b) and (4c), I look forward to the authors' results in the revised manuscript.
>
> Concerning (7a), (7b), and (7c), I suggest the authors explore these aspects in their future work; a robust model should accommodate certain special scenarios. Given the current time constraints, it is recommended that the authors investigate these directions in future research, which I hope will be beneficial.
>
> For (6a), (6b), (6c), (8a), and (8c), a brief mention in the manuscript would suffice; even as concise as in the authors' response. This will significantly enhance the clarity of the paper.
>
> Regarding (8b), (13a), and (14b), these can be presented in future work, as the authors have already conceived of and conducted preliminary research on some of these issues.
>
> Notably, question 2(a) is not out of scope. As the regularization structure constituting the core innovation of the paper, the HSIC bottleneck has only been validated experimentally for its effectiveness. However, the authors merely inferred the monotonic decrease of the empirical objective under SGD based on the qualitative description that the objective function is bounded and differentiable, lacking rigorous mathematical proof to support the necessary and sufficient conditions for its convergence. This limits the accurate assessment of the method's applicability and reliability. I provide possible solution directions herein for reference only: First, based on the authors' defined objective function structure (cross-entropy loss + bounded and differentiable HSIC regularization term), utilize Lipschitz continuity theory to derive gradient properties. Incorporate the classical convergence theory of the SGD optimization framework to prove the monotonicity of the empirical loss and deduce the convergence rate. Second, draw on the generalization error bound analysis framework of the Variational Information Bottleneck (VIB), transform the HSIC metric into a mutual information constraint, and combine the kernel representation of HSIC to decompose the generalization error into empirical error and feature complexity terms, thereby quantifying the bounds of the model's generalization ability. These may contain inaccuracies and are provided solely as directional references, with the hope of offering some assistance for the authors' future research.

---

> ### Author Response · Authors · 2025-11-21
>
> We sincerely thank the reviewer for the very careful reading of our response and for the willingness to raise the score, conditional on a thorough revision. We are grateful for the constructive feedback, which will substantially improve the paper.
>
> For (4b) and (4c), we will incorporate the additional results and analyses into the future revised manuscript. These experiments are nearly completed.
>
> Concerning (7a), (7b), and (7c), we agree with the reviewer that a robust model should ultimately account for such special scenarios. Due to time and space constraints, we will treat these directions as explicit future work and add a discussion section to (i) acknowledge these limitations and (ii) outline how our framework could be extended to handle these more challenging corner cases.
>
> For (6a), (6b), (6c), (8a), and (8c), we will add explicit mentions in the future revised manuscript to enhance the clarity of the paper.
>
> Regarding (8b), (13a), and (14b), we will mention these directions in the discussion section so that readers can see how the proposed HSIC bottleneck could be further generalized and strengthened.
>
> We are particularly grateful for the reviewer’s detailed and insightful comments on Question 2(a). We agree that, as the HSIC bottleneck is a central innovation of the paper, its optimization properties and generalization behavior deserve a more rigorous mathematical treatment. We thank the reviewer for providing a clear direction, and we will definitely look into this in future work.
>
> Once again, we thank the reviewer for the constructive feedback, the concrete theoretical pointers, and the willingness to significantly upgrade the evaluation. We are committed to implementing all the requested revisions and clarifications to ensure that the revised manuscript fully reflects these improvements.

---

> > ### Comment · Reviewer_BxLj · 2025-11-27
> > **RE: Official Comment by Authors**
> >
> > I thank the authors for the clarification and new results. The rebuttal resolves most of concerns about the effects of prompt designs, and mode selection. I would like to keep my original rating, which is already positive.

---

### Official Review · Reviewer_cumQ · 2025-11-01

**Soundness:** 2
**Presentation:** 3
**Contribution:** 3
**Rating:** 6
**Confidence:** 4

**Summary:**

This paper proposes a HSIC-based bottleneck to enhance the generalization of CLIP features for synthetic image detection across diverse generator families. The method encourages representations to retain label-relevant information while suppressing spurious correlations with input semantics, and further introduces HSIC-Guided Replay (HGR) to mitigate catastrophic forgetting in domain-incremental learning. Experiments on diffusion, GAN, and 3D Gaussian Splatting (3DGS) models demonstrate improved cross-generator performance and continual learning stability compared to recent baselines.

**Strengths:**

- The paper articulates the challenge of detectors overfitting to generator-specific artifacts and semantics, which is a timely and relevant problem for synthetic image forensics.
- The continual learning setting further shows practical awareness of evolving generative models, making the work meaningful for long-term applicability.
- The adaptation of HSIC into the CLIP feature pipeline is implemented in a straightforward and principled manner, and the loss formulation is coherent with prior HSIC-based bottleneck approaches.
- Ablation studies on HSIC components, use of intermediate ViT features, and kernel options provide supportive empirical evidence that each design choice is beneficial.
- The evaluation spans distinct generative paradigms, and the method shows consistent gains across them, indicating improved robustness of the learned representations.
- The 3DGS results highlight the method’s applicability to emerging synthetic formats beyond classical image generators.

**Weaknesses:**

- While effective, the core contribution largely builds on established concepts such as HSIC-based information bottlenecks and CLIP feature refinement. The paper does not sufficiently articulate what is fundamentally novel beyond applying HSIC to a different backbone and combining it with a replay mechanism. As a result, the contribution may be perceived as incremental rather than conceptually innovative.
- The paper provides qualitative intuition and t-SNE visualizations but lacks deeper analysis on what specific semantic attributes are suppressed or preserved through HSIC regularization. A more detailed investigation into feature disentanglement, representation shift, or artifact suppression would strengthen interpretability and scientific value. Without such analysis, the method may appear as a black-box regularizer rather than a principled representation intervention.
- The experiments focus on ProGAN, SDv1.4, and 3DGS-based generators, which do not reflect the current state of generative technology, such as Stable Diffusion 3+, Midjourney, Sora 2, or FLUX models. Since newer generators produce more photorealistic and harder-to-detect outputs, evaluation on these models is essential to demonstrate real-world utility. The absence of such results weakens the strength of the claimed “generalization” capability.

**Questions:**

- How does HSIC specifically reshape CLIP features at different semantic granularity levels? Can the authors provide more concrete evidence—beyond t-SNE—that illustrates which types of semantic or generator-specific correlations are suppressed?
- How well does the method scale when extended to modern, highly realistic generators such as SD3.5, Midjourney, Sora2, or FLUX? Have the authors tested whether the model remains effective with these more challenging sources?
- In continual learning scenarios with more than 6–8 sequential domains, does HGR maintain long-term stability? Could the authors provide results over longer task horizons to support claims of scalability and robustness?
- What is the computational overhead of using 24-layer CLIP features + HSIC loss during training and inference? Could the model be made more efficient without sacrificing performance?
- Several recent approaches build upon CLIP to improve cross-generator generalization, and comparing against such methods would better contextualize the contribution of the proposed HSIC bottleneck.

---

> ### Author Response · Authors · 2025-11-20
> **How does HSIC specifically reshape CLIP features at different semantic granularity levels?**
>
> In the revision, we provide a quantitative analysis in Appendix Figure 12 illustrating how the HSIC bottleneck affects text-alignment semantics on the ProGAN test set.
>
> For each ProGAN class label (airplane, bicycle, …, tvmonitor), we compute the cosine similarity between the CLIP image embedding and the CLIP text embedding of the prompt “an image of [class]”. With the original CLIP features, similarities are consistently around **0.18–0.24** across all 20 classes (e.g., boat, bottle, bus, car, cat, chair), indicating strong object-level text alignment even for ProGAN synthetic images.
>
> We then train the model with the HSIC bottleneck (using a 768-dimensional compressed representation for this analysis) and recompute the similarities. They drop to around **0.01** for every class, showing that HSIC strongly suppresses object-class text-alignment semantics while preserving the information needed for the real-vs-synthetic decision.

---

> ### Author Response · Authors · 2025-11-20
> **How well does the method scale when extended to modern, highly realistic generators?**
>
> Our experiments are based on the GenImage benchmark, which is one of the most recent large-scale benchmarks for synthetic image detection and is also used by our main baseline VIB-Net (CVPR 2025). GenImage includes diverse, high-quality diffusion-based generators, and our method consistently outperforms prior work under these settings.
>
> We agree that evaluating on even newer models, such as SD3.5, Midjourney, Sora2, or FLUX, is important for assessing real-world robustness. We are searching for newly released 2025 datasets and will add the results in the future revised version if time permits.

---

> ### Author Response · Authors · 2025-11-20
> **In continual learning scenarios with more than 6–8 sequential domains, does HGR maintain long-term stability?**
>
> In this submission we focus on a 3-domain incremental setting, which already reveals clear differences between HGR and the baselines. Extending the study to longer domain sequences (6–8+ domains) is computationally demanding and is therefore left as an important direction for future work.

---

> ### Author Response · Authors · 2025-11-20
> **What is the computational overhead of using 24-layer CLIP features + HSIC loss during training and inference?**
>
> We train the HSIC bottleneck on pre-extracted CLIP features, so the main extra cost over a standard CLIP linear probe is computing and centering the batch-wise Gram matrices for HSIC. For typical batch sizes (64–1024), this overhead is modest and often negligible in practice, and HSIC is used only during training.
>
> In inference, our model reduces to CLIP features plus a linear encoder and a classifier head, so the cost is essentially the same as a CLIP-based detector.

---

> ### Author Response · Authors · 2025-11-20
> **While effective, the core contribution largely builds on established concepts such as HSIC-based information bottlenecks and CLIP feature refinement. The paper does not sufficiently articulate what is fundamentally novel beyond applying HSIC to a different backbone and combining it with a replay mechanism. As a result, the contribution may be perceived as incremental rather than conceptually innovative.**
>
> Our HSIC bottleneck is indeed related to these works, but it targets a different problem setting and is combined with components that are specific to synthetic image detection. We list three main different points:
>
> 1. **Problem setting and objective.**
>    DualHSIC is designed for supervised class-incremental learning and uses two HSIC terms (bottleneck + alignment) to mitigate inter-task interference (e.g., on Split CIFAR). Our bottleneck instead focuses on synthetic image detection with cross-generator generalization, where the goal is to suppress CLIP’s text-alignment semantics.
>
> 2. **Where HSIC is applied.**
>    DualHSIC applies HSIC to intermediate features of a ResNet, and these features themselves serve as the representation $Z_j$ ($j = 1, \dots, L$) in Section 3.1, Eq. (5). In contrast, we attach a compact HSIC bottleneck on top of a frozen CLIP image encoder: CLIP intermediate features are treated as the input $x$, and a simple linear layer learns the bottleneck representation $z$ in Section 3.1, Eq. (6).
>
> 3. **Usage of CLIP feature.**
>    RINE collects CLS tokens from all intermediate CLIP encoder blocks and then applies the Trainable Importance Estimator (TIE) module to weight each block before aggregating them into a single feature vector per image. In contrast, our HSIC bottleneck simply concatenates the features from all intermediate layers of CLIP and reshapes it via a compact linear layer. Thus, our approach is conceptually different from RINE’s block-aggregation design. The only similarity between our method and RINE is that both leverage features from intermediate CLIP layers.
>
> In the revised version, we have made changes to emphasize the **novelties** of i) **curating a 3DGS synthetic image dataset**, which is particularly challenging for existing synthetic image detectors, and of ii) **providing a continual learning solution tailored to this setting**, which is unexplored by previous synthetic image detection literature.
>
> > To the best of our knowledge, we are the first to introduce a photorealistic 3DGS Synthetic Image Benchmark, together with a continual learning solution tailored to this setting, providing the community with a much-needed, realistic testbed for studying robustness and generalization of synthetic image detectors under evolving 3D generation pipelines.
>
> Here is the link to the test split of GAGAvatar:
> https://osf.io/xwmfj/overview?view_only=af3c697d0c1a470b87961168fc50bcac

---

### Author Response · Authors · 2025-11-20
**Update of a revised PDF**

Dear Reviewers and Area Chairs,

Thank you for the time and effort you have dedicated to reviewing our paper. We greatly appreciate the valuable and insightful feedback provided by the reviewers. We have addressed each comment in the individual reviews and uploaded a revised PDF that includes the following changes:

1. We have emphasized the novelty of curating a 3DGS synthetic image dataset, which is particularly challenging for existing synthetic image detectors, and of providing a continual learning solution tailored to this setting.

2. We have added results where existing detectors are evaluated on 3DGS synthetic images in Tables 3 and 4. This comparison makes the gap to 3DGS explicit and shows that our HGR narrows this gap while maintaining strong performance on the original generators.

3. The related work section has been updated to include DualHSIC, incorporate more recent studies, and clearly position our approach as a replay-based sampling strategy tailored to synthetic image detection.

4. We have revised Sec. 2.3 to explicitly define $\mathbf{1}$ as the all-ones vector, $\mathrm{tr}(\cdot)$ as the standard matrix trace operator, and $\mathbf{I}_n$ as the identity matrix.

5. The implementation details of the encoder and classifier have been included.

6. We have provided additional ablation studies in the Appendix, varying activation functions (ReLU, Sigmoid, Tanh), batch size (64, 128, 256, 512, 1024), hidden dimension (64, 128, 256, 512, 1024, 2048), and number of layers (1–6).

7. We have also provided a quantitative analysis of how the HSIC bottleneck affects text-alignment semantics on the ProGAN test set

8. A sensitivity analysis of HSIC's $\lambda_{x}$ and $\lambda_{y}$ on SDV1.4 and ProGAN has been included.

9. We have added an ablation study of HSIC relevance in HSIC-Guided Replay (HGR).

&#x205F;
#### **Table 7: Ablation of HSIC relevance in HSIC-Guided Replay (HGR) on the SDV1.4 source. Numbers denote (mACC/mAP).**

| Method             | Diffusion       | GANs            | Others          | GHA             | SA              | GAGAvatar       | Average         |
| ------------------ | --------------- | --------------- | --------------- | --------------- | --------------- | --------------- | --------------- |
| w/o HSIC relevance | 95.39/99.80     | **94.15/99.13** | **82.99/94.42** | 94.75/99.62     | **99.06/99.98** | **95.92/99.13** | 93.80/98.94     |
| HGR                | **97.12/99.81** | 94.00/99.07     | 82.31/94.29     | **97.06/99.77** | 98.07/99.99     | 95.18/99.07     | **94.38/98.92** |

&#x205F;
#### **Table 8: Ablation of HSIC relevance in HSIC-Guided Replay (HGR) on the ProGAN source. Numbers denote (mACC/mAP).**

| Method             | Diffusion       | GANs            | Others          | GHA             | SA              | GAGAvatar       | Average         |
| ------------------ | --------------- | --------------- | --------------- | --------------- | --------------- | --------------- | --------------- |
| w/o HSIC relevance | 77.54/93.55     | **94.35/98.89** | 77.95/89.48     | 91.49/99.36     | **99.60/99.99** | **96.31/99.53** | 86.23/95.89     |
| HGR                | **82.87/94.33** | 93.94/98.85     | **80.99/90.72** | **94.71/99.47** | 99.45/100.00    | 95.47/99.26     | **88.63/96.31** |

---

### Author Response · Authors · 2025-11-25
**Update of a second revised PDF**

Dear Reviewers and Area Chairs,

Thank you for the time and effort you have dedicated to reviewing our paper. We greatly appreciate the valuable and insightful feedback provided by the reviewers. We have addressed each comment in the individual reviews and uploaded a second revised PDF that includes the following changes:

1. We reorganized the related work on generalized synthetic image detection to: (1) add more citations on techniques that identify forensic cues for detecting synthetic images and include additional citations on CLIP-based detectors. (2) introduce a dedicated paragraph on diffusion-generated image detection (e.g., DIRE and DRCT); and (3) discuss the synthetic generation models in the Appendix.

2. We clarified our motivation of exploring replay-based method.

> Since replay is arguably the most widely adopted and practically deployed mechanism in continual learning, we focus on what we view as its key factor: the sampling strategy. As the first work to study a continual learning setup for synthetic image detection, we build on this rehearsal-based line of work in the domain-incremental setting and introduce HSIC-Guided Replay, a replay-based sampling strategy tailored to synthetic image detection that scores and selects exemplars to preserve coverage of prior domains while adapting to new ones.

3. We reported the results of our method evaluated on the DRCT-2M benchmark introduced by DRCT [A] in the Appendix. DRCT-2M contains a diverse collection of synthetic images generated from multiple Stable Diffusion families and variants, including LDM, SDv1.x/SDv2, SDXL, refiner and turbo variants, LCM-based accelerations, and ControlNet-based pipelines. Table 10 reports the performance of CNNSpot, F3Net, CLIP/RN50, Conv-B, UnivFD, DIRE, and DRCT from the orignial DRCT paper along with our methods. We follow the same training protocol to train on the real images of MSCOCO and the synthetic images generated by SDV1.4 and then evaluated on different testing subsets on DRCT-2M. While DRCT already outperforms prior baselines in average accuracy, our method (**Ours**) further improves performance across almost all diffusion variants, and the variant that additionally exploits intermediate CLIP features (**Ours w/ intermediate**) achieves near-saturated performance on all generators.

[A] DRCT: Diffusion Reconstruction Contrastive Training towards Universal Detection of Diffusion Generated Images, Chen et al., ICML 2024.

4. We reported RINE results trained on SDV1.4 in Table 1 with official code.

5. We clarified the focus on the 3-domain incremental setting due to time limits, using only 3DGS-rendered synthetic images but no NeRF-rendered due to photorealistic.

6. We mentioned the forward and backward transfer analysis.

7. We added discussion of future work and extensions, including (1) extremely low-resolution inputs (e.g., 32 × 32, 64 × 64) (2) heavily post-processed images (e.g., strong JPEG compression, Gaussian blur) (3) adversarial robustness (e.g., FGSM/PGD attacks) (4) highly imbalanced data (e.g., 1:5, 1:10) (5)  multi-modal inputs (e.g. 3D point clouds) (6) alternative aggregation schemes (e.g., attention-based fusion) (7) lighter models (e.g. MobileNetV3, EfficientNet-B0) (8) uncurated, real-world platforms (e.g. Reddit)

8. We added quantitative clustering metrics (silhouette coefficient, Davies-Bouldin index) in Appendix. We group generators into three families—Diffusion, GANs, and Others (deepfake and SAN)—and report the mean silhouette / DB score for each family in Table 9. The first row (`x`) uses the original CLIP image features, while the second and third rows (“SDV1.4 `z`” and “ProGAN `z`”) use the HSIC bottleneck representations trained on SDV1.4 and ProGAN, respectively.

**Table 9.** Silhouette coefficient (higher is better) and Davies–Bouldin index (lower is better) computed from image embeddings, grouped by generator family (Diffusion, GANs, Others). The first row (`x`) uses the original CLIP image features, while the second and third rows (“SDV1.4 `z`” and “ProGAN `z`”) use HSIC bottleneck representations trained on SDV1.4 and ProGAN, respectively. Best values in each column are bolded.

| Embedding | Training Source | Diffusion        | GANs              | Others             |
|:---------:|:---------------:|:----------------:|:-----------------:|:------------------:|
| `x`       | -               | 0.03 / 5.64      | 0.08 / 3.75       | 0.04 / 5.37        |
| `z`       | SDV1.4          | **0.65 / 0.45**  | 0.59 / 0.53       | 0.30 / 1.40        |
| `z`       | ProGAN          | 0.34 / 1.19      | **0.69 / 0.40**   | **0.34 / 0.98**    |


9. We added Figure 15 to report the mean accuracies and standard deviations of HGR over 5 runs. HGR achieves consistently high accuracy with small variance across runs. This indicates that our improvements are not due to a favorable random seed, and that
HGR yields stable performance under repeated training.

---

### Author Response · Authors · 2025-12-01
**Summary for ACs**

Dear Area Chairs,

We appreciate the time and effort you have dedicated to reviewing our paper. Below we summarize some major questions commonly raised by the reviewers and how we addressed the main concerns.

## 1. Novelty of our method

Reviewers cumQ and zYvc expressed concerns that our HSIC bottleneck is an incremental combination of the HSIC bottleneck in DualHSIC [A] and the CLIP-based detector RINE [B].

In our rebuttal, we clarified that our method targets a different problem setting and uses components tailored specifically to synthetic image detection. We highlighted three key distinctions:

**Problem setting and objective.** DualHSIC is designed for **class-incremental learning**, using two HSIC terms (bottleneck + alignment) to mitigate inter-task interference on benchmarks such as Split CIFAR. In contrast, our bottleneck is developed for **synthetic image detection** with cross-generator generalization, where the central goal is to suppress CLIP’s text-alignment semantics that otherwise hinder robust real-vs-synthetic discrimination.

**Where HSIC is applied.** DualHSIC applies HSIC to intermediate features of a ResNet, and these features themselves serve as the representation $z$ in Section 3.1, Eq. (5). In contrast, we attach a compact HSIC bottleneck on top of a frozen CLIP image encoder: CLIP intermediate features are treated as the input $x$, and a simple linear layer learns the bottleneck representation in Section 3.1, Eq. (6).

**Usage of CLIP feature.** RINE collects CLS tokens from all intermediate CLIP encoder blocks and then applies the Trainable Importance Estimator (TIE) module to weight each block before aggregating them into a single feature vector per image. In contrast, our HSIC bottleneck simply concatenates the features from all intermediate layers of CLIP and reshapes it via a compact linear layer. Thus, our approach is conceptually different from RINE’s block-aggregation design. The only similarity between our method and RINE is that both leverage features from intermediate CLIP layers.

[A] DualHSIC: HSIC-Bottleneck and Alignment for Continual Learning, Wang et al., ICML 2023.

[B] Leveraging Representations from Intermediate Encoder-blocks for Synthetic Image Detection, Koutlis et al., ECCV 2024.

## 2. Results in more recent generators
Reviewers cumQ and 91jn raised concerns about how well our method generalizes to more recent generators. Our main experiments are conducted on the GenImage benchmark [C], one of the most recent large-scale benchmarks for synthetic image detection, and the same benchmark used by our main baseline VIB-Net [D].

We agree that evaluating on newer models is important for assessing real-world robustness. To address this, we additionally reported results of our method on the DRCT-2M benchmark introduced by DRCT [E]; these results are included in **Appendix Table 10**.

[C] GenImage: A Million-Scale Benchmark for Detecting AI-Generated Image, Zhu et al., NeurIPS 2023.

[D] Towards Universal AI-Generated Image Detection by Variational Information Bottleneck Network, Zhang et al., CVPR 2025.

[E] DRCT: Diffusion Reconstruction Contrastive Training towards Universal Detection of Diffusion Generated Images, Chen et al., ICML 2024.

## 3. How does HSIC specifically reshape CLIP features?
Reviewers cumQ and BxLj asked for a quantitative analysis of how the HSIC bottleneck reshapes CLIP features. In response, we added an analysis in **Appendix Figure 12** that illustrates how the HSIC bottleneck affects text-alignment semantics on the ProGAN test set.

For each ProGAN class label (airplane, bicycle, …, tvmonitor), we compute the cosine similarity between the CLIP image embedding and the CLIP text embedding for the prompt “an image of [class]”. Using the original CLIP features, these similarities are consistently around 0.18–0.24 across all 20 classes (e.g., boat, bottle, bus, car, cat, chair), indicating strong object-level text alignment even for ProGAN synthetic images.

We then train the model with the HSIC bottleneck (using a 768-dimensional compressed representation for this analysis) and recompute the similarities. They drop to around 0.01 for every class, demonstrating that the HSIC bottleneck strongly suppresses object-class text-alignment semantics while preserving the information necessary for the real-vs-synthetic decision.

## 4. Ablation of the HSIC relevance term in HGR

We thank reviewers 91jn and zYvc for highlighting the need to isolate the effect of the HSIC relevance term in Eq. (10).
In the revised **Appendix Tables 7 and 8**, we report an ablation where we construct a k-center only variant by removing the HSIC component and setting $\lambda_{kc}$ = 1. The results confirm that the full HGR (with the HSIC relevance term) provides gains beyond standard coverage-based replay.

---

### Author Response · Authors · 2025-12-01
**Summary for ACs**

## Other responses to reviewer zYvc

We appreciate reviewer zYvc's observation that a key strength of our work is the inclusion of a 3DGS synthetic image dataset in the continual learning setup. In the revised version, we have made following changes to highlight our novelty and contribution:

1. **Emphasize the novelty** of curating a 3DGS synthetic image dataset, which is particularly challenging for existing synthetic image detectors, and of providing a continual learning solution tailored to this setting.

> To the best of our knowledge, we are the first to introduce a photorealistic 3DGS Synthetic Image
Benchmark, together with a continual learning solution tailored to this setting, providing the community with a much-needed, realistic testbed for studying robustness and generalization of synthetic
image detectors under evolving 3D generation pipelines.

2. **Add results** where existing detectors are evaluated on 3DGS synthetic images in Tables 3 and 4. This comparison will make the gap to 3DGS explicit and show that our HGR narrows this gap while maintaining strong performance on the original generators.

## Other responses to reviewer 91jn
In the following table we list other concerns raised by reviewer 91jn and how we refined our paper in response.

&#x205F;

| Question            | Response      |
| ------------------ | --------------- |
| In the DIL setting, the comparison methods are out-of-date. | In the current submission, we focused on widely used **replay-based sampling baselines** that can be reliably adapted to our detection-oriented setting. Our contribution is largely orthogonal to most recent DIL advances—which often target improved parameter/optimizer selection, regularization, or architectural changes—because our method specifically proposes an HSIC-guided replay-based sampling strategy.     |
| Given that other works more recently focused on parameter/optimizer selection, regularization, or architectural changes, why did you choose to return to exploring how replay can increase performance? | Since our architecture is fixed—using a CLIP backbone and its intermediate features—we do not explore additional architectural changes for continual learning. Moreover, replay is arguably the most widely adopted and practically deployed mechanism in continual learning, including in methods that primarily present themselves as regularization- or architecture-based. As the first work to study a continual learning setup for synthetic image detection, we therefore focus on what we view as the key factor within the replay mechanism: the sampling strategy. |
| The related work section is not in-depth enough | We made the following updates to the related work section: (1) added an introduction to DualHSIC, (2) included recent diffusion-specific synthetic image detection methods (DIRE and DRCT), (3) clarified the development trajectory of synthetic image detection to form a more complete narrative, and (4) cited additional recent works on synthetic image detection and continual learning. |
| Mathematical background section (2.3) misses explicitly defining some mathematical notation. | We added explicit definition. |

## Discussion with reviewer BxLj
We adopted a point-by-point communication format with reviewer BxLj to ensure that each concern was precisely identified and addressed. We believe that walking through the details of this discussion will help the ACs clearly understand our communication with reviewer BxLj.

---

### Author Response · Authors · 2025-12-02
**Timeline of Evaluation Score Changes for ACs**

At last, we would like to sincerely thank the reviewers for the time and effort dedicated to evaluating our paper. We greatly appreciate the thoughtful and constructive feedback, as well as the willingness of the reviewers to revise their evaluation scores in light of our responses and revisions. Here, we would like to briefly highlight for the ACs when and how these score changes occurred.

After receiving the reminder from reviewer BxLj on **Nov 20**, we released the responses to the reviewers’ questions that we had been preparing since the official reviews were first posted. We are very grateful to reviewer BxLj for the willingness to raising the score. After we addressed the requested revisions, reviewer BxLj raised the evaluation score to 10, before the official reviews were reverted. As stated in the comment:

> “After carefully reviewing the authors' response, I intend to raise my evaluation score to 10 on the condition that the authors have thoroughly revised the manuscript as requested.”

For reviewer 91jn, we carefully revised many of the points raised, including: improving the completeness of related work and citations, clarifying our motivation for exploring replay-based sampling methods, refining the mathematical notations, and — most importantly — adding results on more recent datasets. Reviewer 91jn explicitly noted:

> “Thank you for looking into this. I believe it would be a strong addition to your paper and the factor most likely to incline me to raise the score.”

and later:

> “I would like to thank the authors for all their efforts in the discussion of this paper, I view it much more favorably after the discussion and have raised my score accordingly.”

We are very thankful to reviewer 91jn for pointing out the incompleteness of our original submission, which helped us improve the paper substantially. Reviewer 91jn raised the evaluation score to 6 on **Nov 26**.

For reviewer zYvc, we greatly appreciate the recognition of our strengths, particularly the inclusion of a 3DGS synthetic image dataset in a continual-learning setting, which allowed us to better clarify our novelty and contribution in the introduction. After our clarifications, reviewer zYvc commented:

> “The authors have satisfactorily addressed my questions and the weaknesses raised. I am increasing my score to 6.”

Reviewer zYvc raised the evaluation score to 6 on **Nov 26**.

We kindly emphasize these dates because all of them occurred before the widespread dissemination of information about the OpenReview/ICLR 2026 leakage issue and before the official statement from the ICLR 2026 organizers on **Nov 27**. We appreciate the Area Chairs considering this timing for our submission. Thank you for your understanding.

---

### Meta-Review · Area_Chair_rsZi · 2026-01-07

**Summary:**

The manuscript received divergent ratings.

Reviewer cumQ (rating 6) requested clearer evidence of how HSIC reshapes CLIP features, evaluation on other generators, analysis of computational overhead, and comparisons with recent CLIP-based methods. Also, the novelty issue is questioned.

Reviewer BxLj (rating 4) gave a very comprehensive review covering various aspects of this paper, including the experiments, the presentation, and the rationale, etc.

Reviewer 91jn (rating 2) argues that the comparisons and the chosen models are outdated, related work is insufficiently current, mathematical notation lacks clarity, and key ablations (replay choice) are missing.

Reviewer zYvc (rating 4) mainly questioned the novelty, but acknowledged the contribution of 3DGS samples in this field.

**Reviewer Concerns:**

Reviewer BxLj has successfully engaged in the discussion with the authors and was generally satisfied with the rebuttal. A remaining issue is Question 2(a), which the authors claim that a formal analysis of the HSIC bottleneck's convergence is out of the scope, while Reviewer BxLj did not agree.

Reviewer 91jn and Reviewer zYvc are also convinced by the rebuttal, and claim to increase the score in the response.

**Reviewer Scores:**

Reviewer cumQ will keep the score 6, and the other three reviewers will increase the ratings to 6.

---

### Decision · Program_Chairs · 2026-01-26

Accept (Poster)